# The soil-borne white root rot pathogen *Rosellinia necatrix* expresses antimicrobial proteins during host colonization

Edgar A. Chavarro-Carrero[1,2], Nick C. Snelders[2,3ʘ], David E. Torres[1,3ʘ], Anton Kraege[2], Ana López-Moral[2], Gabriella C. Petti[2], Wilko Punt[2], Jan Wieneke[2], Rómulo García-Velasco[4], Carlos J. López-Herrera[5], Michael F. Seidl[3‡], Bart P. H. J. Thomma[1,2‡]*

**1** Laboratory of Phytopathology, Wageningen University & Research, Wageningen, The Netherlands, **2** Institute for Plant Sciences, Cluster of Excellence on Plant Sciences (CEPLAS), University of Cologne, Cologne, Germany, **3** Theoretical Biology & Bioinformatics Group, Department of Biology, Utrecht University, Utrecht, The Netherlands, **4** Laboratory of Phytopathology, Tenancingo University Center, Autonomous University of the State of Mexico, Tenancingo, State of Mexico, Mexico, **5** CSIC, Instituto de Agricultura Sostenible, Dept. Protección de Cultivos, C/Alameda del Obispo s/n, Córdoba, Spain

ʘ These authors contributed equally to this work.
‡ MFS and BPT also contributed equally to this work.
* bthomma@uni-koeln.de

**Data Availability Statement:** Genomes and genome annotation data have been submitted to NCBI under BioProject PRJNA727191.

## Abstract

*Rosellinia necatrix* is a prevalent soil-borne plant-pathogenic fungus that is the causal agent of white root rot disease in a broad range of host plants. The limited availability of genomic resources for *R. necatrix* has complicated a thorough understanding of its infection biology. Here, we sequenced nine *R. necatrix* strains with Oxford Nanopore sequencing technology, and with DNA proximity ligation we generated a gapless assembly of one of the genomes into ten chromosomes. Whereas many filamentous pathogens display a so-called two-speed genome with more dynamic and more conserved compartments, the *R. necatrix* genome does not display such genome compartmentalization. It has recently been proposed that fungal plant pathogens may employ effectors with antimicrobial activity to manipulate the host microbiota to promote infection. In the predicted secretome of *R. necatrix*, 26 putative antimicrobial effector proteins were identified, nine of which are expressed during plant colonization. Two of the candidates were tested, both of which were found to possess selective antimicrobial activity. Intriguingly, some of the inhibited bacteria are antagonists of *R. necatrix* growth *in vitro* and can alleviate *R. necatrix* infection on cotton plants. Collectively, our data show that *R. necatrix* encodes antimicrobials that are expressed during host colonization and that may contribute to modulation of host-associated microbiota to stimulate disease development.

## Author summary

Most if not all organisms, including plants, associate with a wide diversity of microbes that live either inside these organisms, or in their immediate vicinity, and that collectively

**Funding:** EACC and DET acknowledge receipt of PhD fellowships from CONACyT, Mexico. ALM is holder of a postdoctoral research fellow funded by the 'Fundación Ramón Areces'. BPHJT acknowledges funding by the Alexander von Humboldt Foundation in the framework of an Alexander von Humboldt Professorship endowed by the German Federal Ministry of Education and Research, and is furthermore supported by the Deutsche Forschungsgemeinschaft (DFG, German Research Foundation) under Germany´s Excellence Strategy – EXC 2048/1 – Project ID: 390686111. The funders had no role in study design, data collection and analysis, decision to publish, or preparation of the manuscript.

**Competing interests:** The authors have declared that no competing interests exist.

form their microbiota. Moreover, increasing evidence reveals that microbiota represent a key determinant for their health. To cause disease on their hosts, microbial pathogens need to overcome host immunity. Conceivably, pathogens need to overcome beneficial contributions that microbiota make to an organism's health. Here, we show that the genome of the fungal white root rot pathogen *Rosellinia necatrix* encodes putatively secreted antimicrobial proteins, many of which are expressed during plant colonization. Two of these proteins are functionally analyzed in this study, and we reveal that they can inhibit the growth of bacteria that antagonise *R. necatrix* growth *in vitro* and that can alleviate *R. necatrix* infection on cotton plants. Thus, we propose that *R. necatrix* employs antimicrobials during host colonization to promote host infection through the selective manipulation of host microbiota.

## Introduction

*Rosellinia necatrix* is a prevalent soil-borne plant-pathogenic fungus that is found in temperate and tropical areas worldwide [1]. As causal agent of white root rot disease, *R. necatrix* has a broad host range comprising at least 170 species of dicotyledonous angiosperms that are dispersed over 63 genera and 30 families [2]. Many of these species are of great economic importance, such as *Coffea* spp. (coffee), *Malus* spp. (apple), *Olea europea* L. (olive), *Persea americana* Mill. (avocado), *Prunus* spp. (peaches, almonds, etc.), *Vitis vinifera* L. (grape) [3] and *Rosa* sp. (rose) [4].

Plants infected by *R. necatrix* usually display two types of symptoms. The first type is displayed below-ground on the root system, where white and black colonies of mycelium can occur on the surface of infected roots. As the fungus penetrates and colonizes the root tissue, the roots acquire a dark brown color [5]. The second type of symptom occurs above-ground. These symptoms can develop rapidly as a consequence of the damaged root system and comprise wilting of leaves, typically after a period of drought or physiological stress, which affects plant vigor and eventually can lead to plant death. Symptoms of *R. necatrix* infection can also appear slowly, leading to a decline in growth, decreasing leaf numbers, along with wilting of leaves, chlorosis, and death of twigs and branches. On perennials these symptoms aggravate over time, and when moisture and temperature are unfavorable, the plant eventually dies [5].

Various studies have addressed the biology of *R. necatrix* [6,7]. Nevertheless, the molecular mechanisms underlying pathogenicity of *R. necatrix* remain largely unexplored, mainly because the limited availability of genomic resources to date complicates investigations into the molecular biology of *R. necatrix* infections. Until recently, only a single Illumina technology short-read sequencing-based *R. necatrix* draft genome assembly was generated of strain W97 that was isolated from apple in Japan [8]. This 44 Mb genome assembly is highly fragmented as it comprises 1,209 contigs with 12,444 predicted protein-coding genes [8]. Recently, another assembly was released of a South-African strain (CMW50482) that was isolated from avocado, yet this assembly is even more fragmented with 1,362 contigs [9].

It is generally accepted that plant pathogens secrete dozens to hundreds of so-called effectors into their host plants to stimulate host colonization [10,11]. Effectors can be defined as small, secreted proteins of ≤300 amino acids that are cysteine-rich and have tertiary structures that are stabilized by disulfide bridges and that are secreted by pathogens to promote disease on plant hosts [12–15]. However, also larger secreted proteins have been found to act as effectors, such as several LysM effectors [16,17]. Furthermore, also non-proteinaceous effectors have been described, such as fungal secondary metabolites as well as small RNA (sRNA)

molecules that are delivered into host cells to suppress host immunity [18]. Many effectors have been shown to act in suppression of host immune responses [10]. However, recent studies have uncovered additional functions of pathogen effector proteins in host colonization [11]. For example, the soil-borne pathogen *Verticillium dahliae* secretes effector proteins to modulate microbiota compositions inside the host plant as well as outside the host in the soil to support host colonization [19–22]. One of them is the effector protein VdAve1 that was shown to display selective antimicrobial activity and facilitate colonization of tomato and cotton plants through host microbiota manipulation by suppressing antagonistic bacteria, such as members of the *Spingomonadaceae* [19]. Besides VdAve1, ten additional potentially antimicrobial effector protein candidates were predicted by mining the *V. dahliae* secretome for structural homologues of known antimicrobial proteins (AMPs) [19]. One of these candidates, VdAMP2, was subsequently found to contribute to *V. dahliae* survival in soil through its efficacy in microbial competition [19]. Another candidate, VdAMP3, was shown to promote microsclerotia formation in decaying host tissue through its antifungal activity directed against fungal niche competitors [20]. Most recently, a multiallelic gene homologous to *VdAve1* was identified as *VdAve1-like* (*VdAve1L*) of which the *VdAve1L2* variant was shown to encode an effector that promotes tomato colonization through suppression of antagonistic Actinobacteria in the host microbiota [22]. These findings have led to the hypothesis that microbiota manipulation is a general virulence strategy of plant pathogens to promote host colonization [21]. However, thus far evidence for such activity has been lacking for other fungal plant pathogens.

Similar to *V. dahliae*, also *R. necatrix* is a soil-borne pathogen that spends at least part of its life cycle in the soil, known to be an extremely competitive and microbe-rich environment [23]. In the present study, we aimed to generate a high-quality genome assembly to mine the *R. necatrix* genome for potential antimicrobial proteins that are exploited during host colonization and test the hypothesis that other pathogenic fungi besides *V. dahliae* exploit effector proteins with antimicrobial activity during host colonization as well.

## Results

### A chromosome-level assembly of *Rosellinia necatrix* strain R18

Thus far, two fragmented *R. necatrix* draft genome assemblies that comprise over 1,200 contigs are publicly available [8,9]. To generate additional and more contiguous *R. necatrix* genome assemblies, we sequenced nine strains of *R. necatrix* that were collected from rose and avocado in Mexico and Spain, respectively, with Oxford Nanopore sequencing technology (ONT). This resulted in genome assemblies of 28 (for strain R18) to 399 (for strain CH12) contigs (Table 1). The most contiguous assembly was obtained for strain R18, isolated from infected rose plants in Mexico.

To further improve the genome assembly of strain R18, additional ONT sequencing was performed using ultra HMW (UHMW) DNA as template. Additionally, chromosome conformation capture (Hi-C) followed by high-throughput sequencing was performed [24,25]. Based on these orthogonal sequencing data, we ultimately assembled the R18 genome into 11 contigs (Table 1). Using Tapestry [26] twenty telomeric regions ([TTAGGG]$n$) were identified (Fig 1A), and ten of the 11 contigs were found to contain telomeric regions at both ends, suggesting that these represent completely assembled chromosomes, which could be confirmed by Hi-C analysis (Fig 1B). The 11[th], and smallest, contig contained no telomeric regions, displayed a markedly higher coverage (read depth) (Fig 1B), and BLAST analysis revealed hits to fungal mitochondrial genomes, showing that this contig belongs to the mitochondrial genome. Accordingly, no mitochondrial genes could be found on any of the ten chromosomal contigs. Thus, we conclude that our assembly contains ten complete chromosomes that compose the

**Table 1. Sequencing summary and genome assembly statistics for nine *Rosellinia necatrix* strains.**

| Strain name | Rn19[a] | CH12[a] | Rn400[a] | R10[a] | R25[a] | R27[a] | R28[a] | R30[a] | R18[a] | R18[b] |
|---|---|---|---|---|---|---|---|---|---|---|
| Isolated from | Avocado | Avocado | Avocado | Rose | Rose | Rose | Rose | Rose | Rose | |
| Country of origin | Spain | Granada, Spain | Granada, Spain | State of Mexico, Mexico | State of Mexico, Mexico | State of Mexico, Mexico | State of Mexico, Mexico | State of Mexico, Mexico | State of Mexico, Mexico | |
| Year | 2001 | 1988 | 1991 | 2005 | 2008 | 2005 | 2011 | 2014 | 2014 | |
| Read N50 (kb) | 23 | 17 | 21 | 18 | 16 | 21 | 21 | 9 | 18 | **30** |
| Coverage | 45X | 46X | 35X | 35X | 45X | 46X | 41X | 52X | 42X | **120X** |
| No. of contigs | 35 | 399 | 37 | 50 | 58 | 31 | 47 | 130 | 28 | **11** |
| Total length (Mb) | 48.2 | 49.9 | 48.2 | 48.9 | 49.2 | 49.2 | 48.7 | 48.3 | 49 | **49.1** |
| Average length (Mb) | 1.3 | 0.1 | 1.3 | 1 | 0.8 | 1.5 | 1 | 0.3 | 0.8 | **4** |
| Maximum length (Mb) | 6.4 | 6.3 | 5.8 | 4.4 | 4 | 6.4 | 6.4 | 4.5 | 4 | **7.1** |
| N50 (Mb) | 3.4 | 4.6 | 3.8 | 1.9 | 2.5 | 3.2 | 3.4 | 1.6 | 3.1 | **5.1** |
| N90 (Mb) | 0.8 | 3.1 | 1.3 | 0.4 | 0.5 | 1.2 | 0.8 | 0.3 | 0.6 | **3.3** |
| GC (%) | 46.28 | 46.37 | 46.37 | 45.35 | 45.49 | 45.43 | 45.50 | 45.99 | 45.35 | **45.46** |
| BUSCO (%) | 76.7 | 69.9 | 70.2 | 68.4 | 78.9 | 72.3 | 68.2 | 80.7 | 82.5 | **97** |

[a]*de novo* assembly based on Nanopore sequencing of high-molecular weight (HMW) DNA.

[b]*de novo* assembly based on Nanopore sequencing of HMW and UHMW DNA and Hi-C sequencing.

nuclear genome, and an 11[th] contig that composes the mitochondrial genome, and thus that we generated a gapless, telomere-to-telomere, genome assembly. To measure the completeness of the genome assembly, we used BUSCO v4 with the Ascomycota dataset [27], which resulted in a completeness score of 97%.

To further explore the *R. necatrix* strain R18 assembly, genome annotation was performed using Funannotate [28] guided by publicly available RNAseq data [8], resulting in 11,760 predicted protein-coding genes (Table 2). Furthermore, effector genes were predicted using EffectorP [29]. The assembled genome of *R. necatrix* strain R18 was predicted to encode 1,126 secreted proteins, of which 192 were predicted to be effector candidates. BLAST searches revealed that 182 of these have homology to Ascomycete fungal proteins, of which 69 had functional domain annotations, while the remaining 113 were annotated as hypothetical proteins (S1 Table). The largest group of annotated effectors were cell wall degrading enzymes, including glycoside hydrolases and carbohydrate esterases that are generally involved in plant cell wall degradation [30,31]. Effectors carrying carbohydrate-binding domains while lacking enzymatic domains, specifically a concanavalin A-like lectin and two cell wall integrity and stress response component (WSC) domain-containing proteins, were also found. Concanavalin A-like lectin has been described as a virulence factor, although no molecular mechanism has been presented [32], while WSC domain-containing proteins can protect fungal cell wall through structural strengthening, but their contribution to plant colonization remains unclear [33]. Additionally, two lysin motif (LysM) effectors were found that typically bind chitin and play roles in virulence through suppression of chitin-triggered immunity or shielding hyphae against host-derived chitinases [34–37]. Four genes were found to encode hydrophobins; proteins that have been implicated in virulence through acting as phytotoxin [38], or promoting attachment to plant surfaces and appressoria formation [39], although negative impacts on host colonization have also been reported [40]. Finally, three potential toxins were found, namely a cerato-platanin domain-containing protein [41] and two necrosis and ethylene-

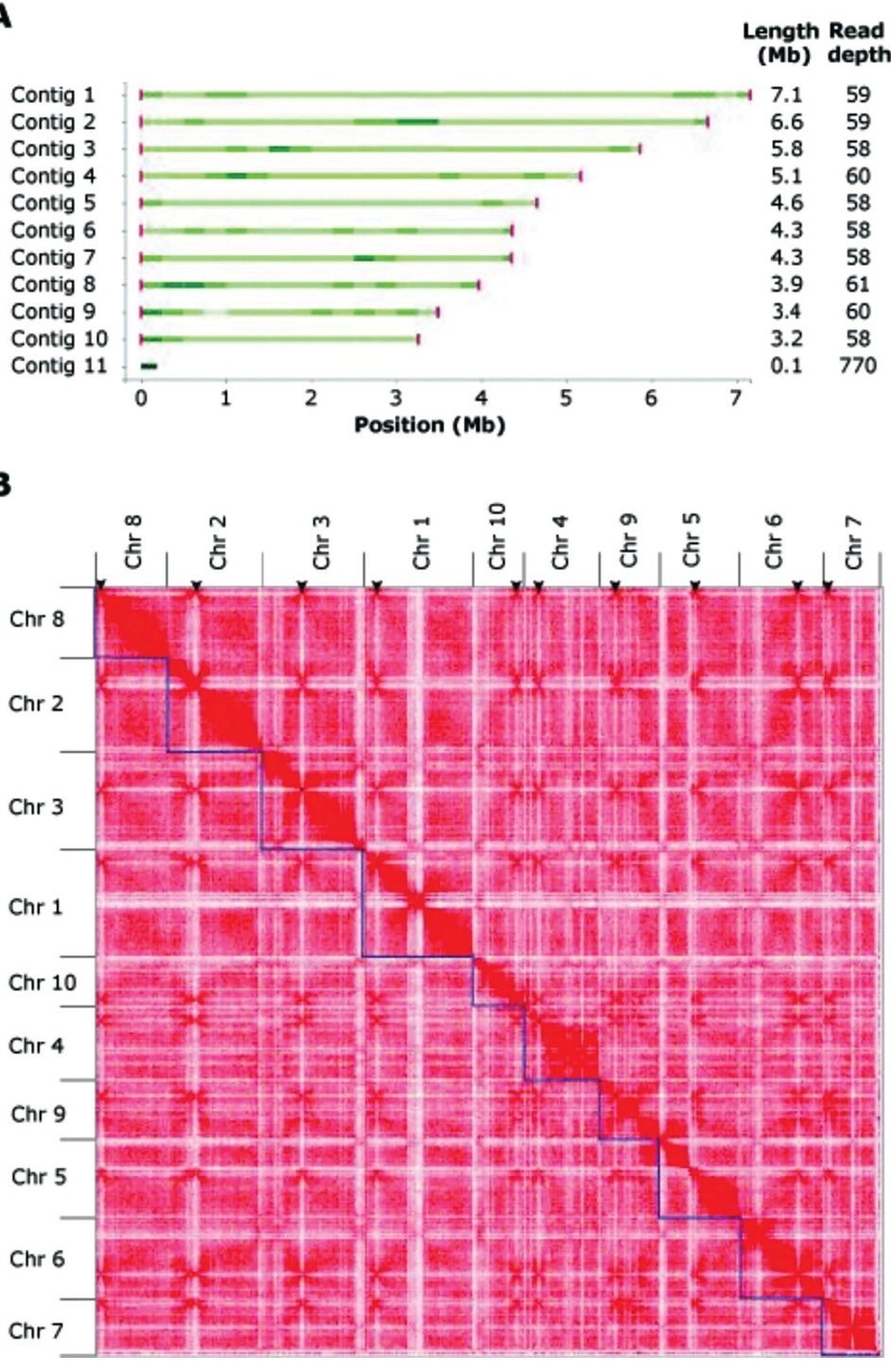

**Fig 1. Genome assembly of *Rosellinia necatrix* strain R18.** (A) Genome assembly plot generated with Tapestry [26]. Green bars indicate contig sizes with the color intensity correlating with read coverage, while red bars indicate telomeres with the color intensity correlating with the number of telomeric repeats. Contig 11 lacks telomeres and represents the mitochondrial genome. (B) Hi-C contact matrix showing interaction frequencies between regions in the genome of *Rosellinia necatrix* strain R18. Ten regions with high inter-chromosomal interaction frequencies are indicative of centromeres and are indicated with arrowheads.

**Table 2. Genome annotation of Rosellinia necatrix strain R18.**

| Genomic trait | Value |
|---|---|
| Protein-coding genes (Funannotate version 1.8.13, [28]) | 11,760 |
| Mean gene length (bp) | 1,689.48 |
| Mean exons per gene | 2.92 |
| Mean intergenic length (bp) | 1,114 |
| Homologs in InterPro database | 8,125 |
| Secreted proteins (SignalP; version 5.0, [44]) | 1,126 |
| Effectors (EffectorP; version 2.0,[29]) | 192 |
| CAZymes (Funannotate) | 535 |
| CAZymes (secreted) (Funannotate) | 300 |
| Small cysteine-rich proteins (Funannotate) | 209 |
| LysM (lysin motif) effectors (Funannotate) | 2 |
| NLP (necrosis and ethylene-inducing-like) effectors (Funannotate) | 2 |
| Secondary metabolite clusters (antiSMASH; version 5.0, [45]) | 48 |

inducing-like (NLP) effectors that are typically thought to be phytotoxic to eudicot plants [42,43] (S1 Table).

To investigate the conservation of effector gene catalogues among the *R. necatrix* genomes that we sequenced in our study, we performed BLAST searches to find homologs of each of the predicted effector genes in the genome of strain R18 in each of the other genomes. Intriguingly, we found remarkably little presence-absence variation, with only few effector genes missing in some of the other genomes. While strains R27, R28 and R30 carry all the effector genes that were identified in strain R18, strains R10 and R25 lack two, strain R19 lacks three, and strains CH12 and Rn400 lack four of the 192 predicted effector genes (S2 Table). Furthermore, we determined the ratio of single nucleotide polymorphisms (SNPs) for each of the predicted effectors. Interestingly, while strains Rn19 and strain Rn400 share no identical effector genes and strain CH12 only a single identical effector gene with strain R18, on the other end of the spectrum strain R10 shares 152 identical effector genes with strain R18. Furthermore, SNP ratios vary from 0% up to 26,12% for effector *FUN_011522* in strain CH12. Generally, we observed higher variability in effectors of the Spanish strains (Rn19, CH12, Rn400) than in the Mexican strains (R10, R25, R27, R28, R30) (S2 Table).

Besides proteinaceous effectors, 48 secondary metabolite gene clusters were predicted using antiSMASH fungal version [45] (Table 2). From these, 13 showed homology to known secondary metabolite clusters, including those for the production of swainsonine, cytochalasin E/K, pyriculol, enniatin, and naphthalene, which have been implicated in pathogenicity through their phytotoxic activity [46–52], and copalyl diphosphate which has been implicated in elongation disorders in plants [53,54] (S3 Table).

## Absence of distinctive genome compartmentalization

In many filamentous pathogens, effector genes are found in repeat-rich and gene sparse genomic compartments, whereas they are depleted in repeat-poor and gene-dense regions that typically harbor housekeeping genes, a genome organization that is typically referred to as the two-speed genome [55]. Thus, we aimed to explore if effector genes were associated with repetitive regions in *R. necatrix* too. Interestingly, the repeat content in *R. necatrix* strain R18 is low (2.39%) and evenly distributed throughout the genome, with the most abundant families being DNA/Helitron (67%) and long terminal repeats (LTRs; 31%). Accordingly, most repetitive elements are not preferentially located near effectors (p>0.05 after permutation test for distance).

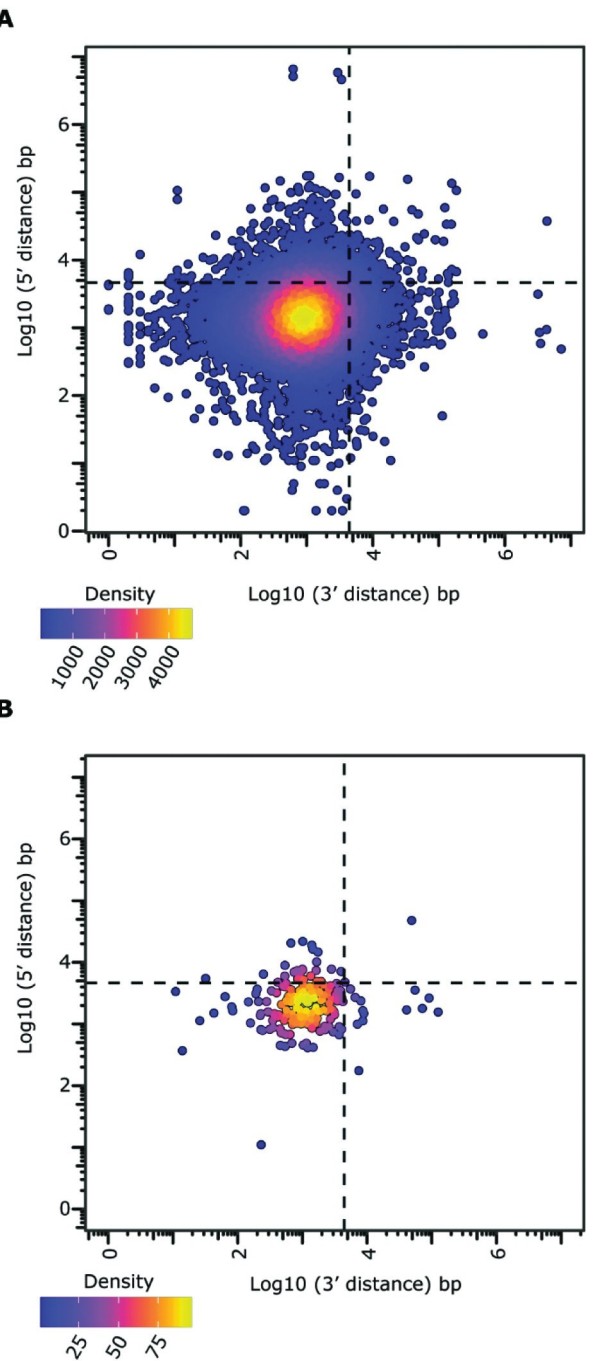

**Fig 2. Effector genes do not localize in gene-sparse regions.** Gene density plot of '5 and '3 flanking log10 transformed intergenic distance with the dashed lines depicting the mean intergenic distances for all genes (A) and candidate effector genes encoded in the *R. necatrix* genome (B).

Moreover, effector genes are not preferentially found in regions with large intergenic distances (Fig 2). Thus, our results do not support association of effector genes with repeat-rich regions in *R. necatrix*.

Genomic comparisons in various filamentous fungi have revealed extensive chromosomal rearrangements and structural variants (SVs) in close association with effector genes [56–60].

Thus, we tested whether effector genes are associated with SVs in *R. necatrix*. First, to explore the genomic diversity among *R. necatrix* strains, a phylogenetic tree of all genomes was constructed with Realphy (version 1.12) [61]. Two clusters were identified, one with strains from Mexico isolated from roses, and one cluster with the strains from Spain and South Africa, isolated from avocado, as well as Japan, isolated from apple (Fig 3). Then, SVs were predicted for each of the genomes sequenced in this study using NanoSV (version 1.2.4 with default settings; [62]) by identifying split- and gapped-aligned long reads for the various strains to define breakpoint-junctions of structural variations when using the gapless genome assembly of strain R18 as a reference. We retrieved 2,639 SVs in total, comprising 1,264 insertions, 1,344 deletions, 4 inversions and 27 translocations (Fig 3A). The number of SVs per strain corresponds to their phylogenetic relationships based on whole-genome comparisons. To investigate the occurrence of the SVs in the *R. necatrix* strains, we calculated the frequency of each SV over the strains used in the analysis. Most of the SVs (94.2%) are shared by <50% of the strains, meaning that they occur in less than four strains. This suggests that SV is a common phenomenon in line with the phylogenetic and geographic relationship of *R. necatrix* strains. Interestingly, SVs occur all across the genome (Fig 3B), largely independently of repetitive regions (p>0.05 after permutation test for distance), but are found in close association with effector genes (p<0.05 after permutation test for distance). Collectively, our results for *R. necatrix* substantiate the lack of the typical genome compartmentalization that is associated with the two-speed genome organization that was found in many other filamentous fungi [63].

## Weak structural clustering in the effector catalog

Effectors continuously evolve towards optimal functionality while simultaneously evading recognition by plant immune receptors, which is considered one of the reasons for the lack of sequence conservation among effector proteins. Despite this lack of sequence conservation, groups of effector proteins that share their three-dimensional structure have been identified in various filamentous phytopathogens, such as the MAX-effector (*Magnaporthe* Avrs and ToxB like) family of the rice blast fungus *Magnaporthe oryzae* [65] that, together with some other families, are also found in the scab fungus *Venturia inaequalis* [66]. It has been speculated that MAX effectors exhibit this structural conservation as an adaptation either to the apoplastic environment, or for transport into the plant cytosol [65]. To investigate whether subsets of *R. necatrix* effectors display a similar fold conservation, their structures were predicted with Alphafold2 with an average quality score in a so-called predicted local distance difference test (pLDDT) of 86.1 (SD 11.8) on a scale from 0 to 100, with 100 indicating a perfect prediction. As only 11% of the predicted structures scored lower than 70 and only 2% lower than 50, we conclude that the fold prediction of the *R. necatrix* effector catalogue is generally robust. Next, we performed similarity clustering of the predicted effector folds, revealing that almost 40% of the effector candidates are structurally unique, while the remaining effectors could be assigned to a total of 31 clusters. As these clusters were only small, with on average only four members, we conclude that relatively little clustering occurs among *R. necatrix* effectors.

To examine whether the observed clustering is merely based on structural similarity, or is mainly driven by sequence conservation, the five largest clusters that contain six to 11 members were further analysed (Fig 4B and 4C). Two of the five clusters show high average template modelling (TM) scores, with 0.85 and 0.91, respectively, on a scale from 0–1, while also showing a relatively high degree of sequence conservation of 43 and 47 percent, respectively. The other three clusters exhibit lower sequence conservation, with maximum 21 percent, but also show considerably lower respectively structural conservation, with TM scores between 0.57 and 0.64 only, indicating that structural similarity within the clusters is positively

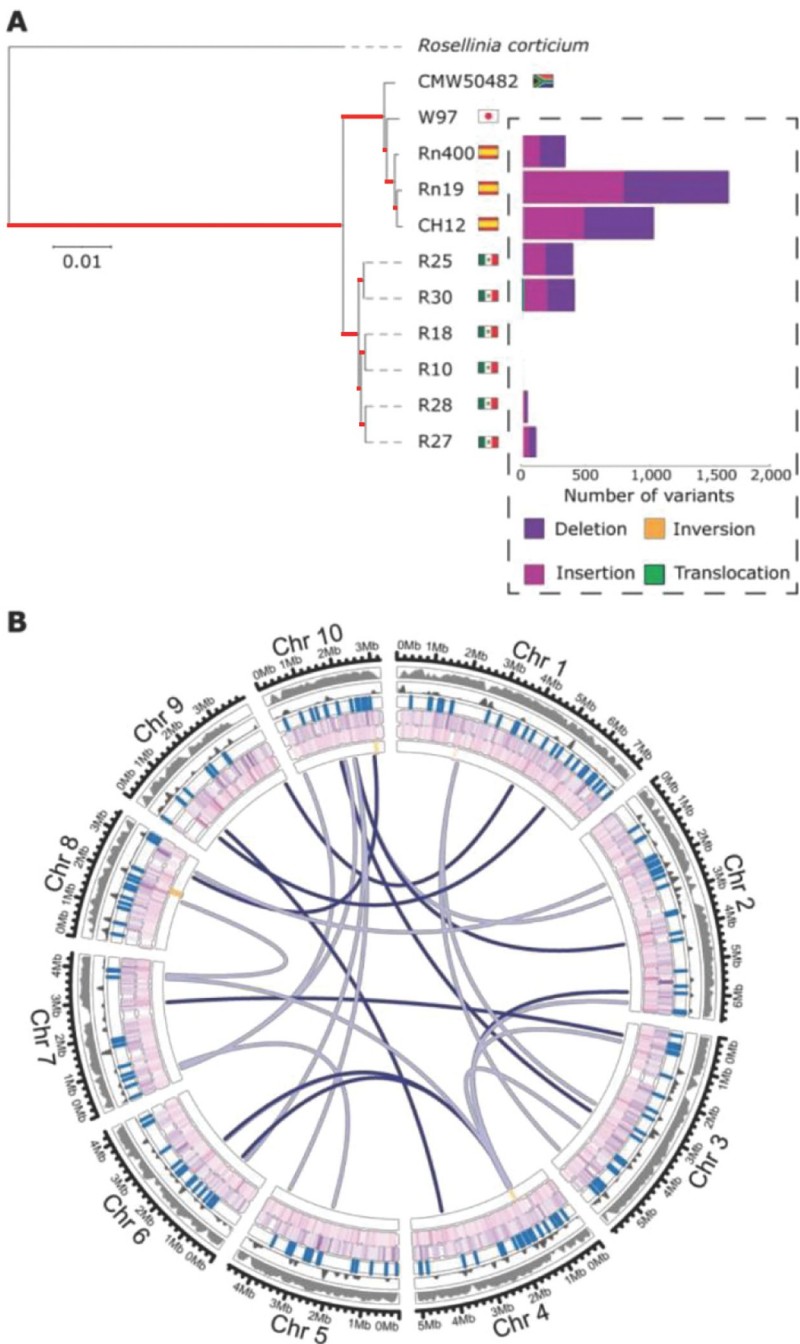

**Fig 3. Phylogeny and structural variation of nine *Rosellinia necatrix* strains.** (A) Phylogeny of sequenced *R. necatrix* strains was inferred using Realphy [64]. The robustness of the phylogeny was assessed using 1000 bootstrap replicates, and branch lengths represent sequence divergence (branches with maximum (100%) bootstrap support are indicated in red). *Rosellinia corticium* was used to root the tree. The amount of structural variants was calculated using NanoSV [62] using strain R18 as a reference. (B) Circular plot displaying the genomic distribution of 2,019 SVs along the 10 chromosomes of *R. necatrix*. The tracks are shown in the following order from outside to inside: Gene density (10 kb), repetitive density (10 kb), effector gene locations, deletions, insertions and inversions and translocations. The color intensity of the lines for each SV track depict frequency in the *R. necatrix* collection.

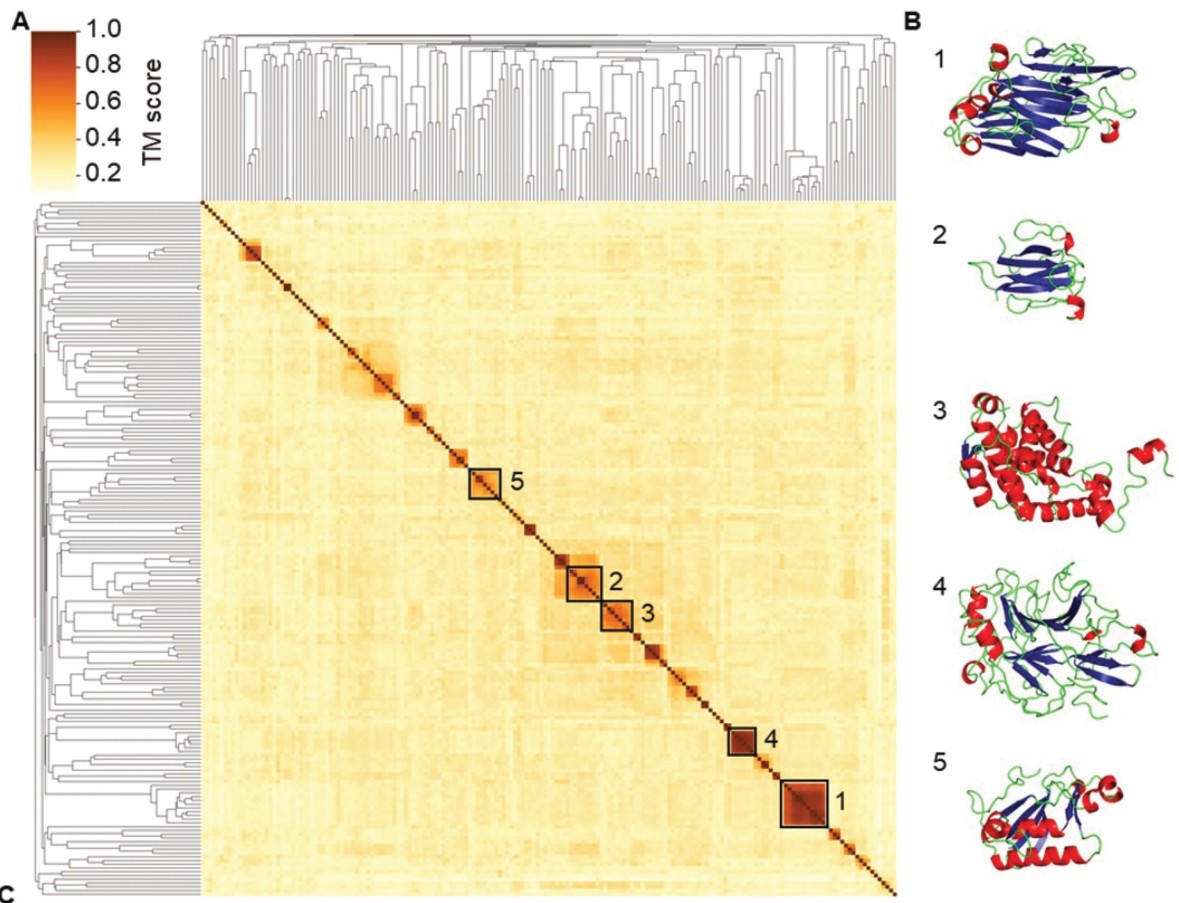

**Fig 4. *R. necatrix* effector candidates show weak structural clustering, which is based on sequence conservation.** (A) Ordered by hierarchical clustering based on structural similarity, the heatmap displays the structural similarity of each effector pair in an all-vs-all alignment based on template modelling (TM) scores that range from 0 to 1. Effector clusters were identified based on a similarity threshold > 0.5. The five largest clusters are highlighted with a black square and ordered by size. (B) Example structure for each of the five clusters based on the effector with the highest similarity to other effectors in the cluster. (C) Characteristics of the five largest structural effector clusters.

| Cluster | Effector proteins | Average TM-score | Average sequence identity ±SD | Annotation |
|---|---|---|---|---|
| 1 | 11 | 0,85 | 43,47 % ±13 | Dehydrogenases |
| 2 | 8 | 0,63 | 17,46 % ±10 | Glycoside hydrolases |
| 3 | 7 | 0,64 | 21,27 % ±4 | Small secreted proteins |
| 4 | 6 | 0,91 | 46,71 % ±5 | Cyanamide hydratases |
| 5 | 6 | 0,57 | 18,29 % ±6 | Glucanosyl-transferases |

correlating with sequence conservation. Hence, the structural clustering can be explained by the sequence conservation amongst the effector proteins. Taken together, although a considerable number of structural effector clusters can be observed in *R. necatrix*, they contain only few members and structural conservation is mostly based on sequence conservation. Thus, effector family expansion seems to have played only a minor role in *R. necatrix* effector evolution.

## Antimicrobial activity in culture filtrates

It has recently been proposed that fungal plant pathogens employ effectors with antimicrobial activity to manipulate the host microbiota to promote infection [10,11,21]. To explore if the soil-borne fungus *R. necatrix* potentially exploits antimicrobials, we first tested whether *R. necatrix* culture filtrates can inhibit the growth of plant-associated bacteria. To this end, we collected *R. necatrix* culture medium after 4, 7 and 9 days of fungal growth by filtration through a 0.45 nm filter, and an aliquot of each of the culture filtrates was heat-inactivated at 95˚C for 10 minutes. Finally, the culture filtrates were used as growth medium for individual bacterial species from a diversity panel of 37 plant-associated bacteria (S4 Table). After overnight incubation, growth of four of the 37 bacteria was inhibited in the 4- and 7-day culture medium filtrates when compared with cultivation in the heat-inactivated culture filtrates, namely *Bacillus drentensis*, *Achromobacter denitrificans*, *Sphingobium mellinum* and *Flavobacterium hauense*. While three of these were not found to be inhibited anymore in the 9-day culture medium filtrate, growth of *B. drentensis* was also still inhibited in this filtrate (Fig 5). Given that the heat-treatment inactivated the antimicrobial activity, suggesting that the activity is of proteinaceous nature, we passed the culture filtrates through a spin column with 3 kDa cut-off and tested the growth of *Bacillus drentensis* and *Flavobacterium hauense* in these filtrates. Interestingly, none of the filtrates inhibited bacterial growth, suggesting that the activity is mediated by proteins that are retained in the spin column, as metabolites and other small molecules are expected to pass through. Collectively, these findings suggest that the culture medium of *R. necatrix* contains (a) heat-sensitive protein(s) with selective antimicrobial activity.

## Prediction of potential antimicrobial effector candidates

We previously showed that structural prediction successfully identified antimicrobial effector proteins encoded in the genome of *V. dahliae* [19,20]. Thus, we similarly mined the secretome of the *R. necatrix* strain R18 for structural homologues of known antimicrobial proteins using Phyre2 [17,19], leading to the identification of 26 candidates with potential antimicrobial activity (Table 3). Subsequently, we used publicly available RNAseq data [8] to assess expression of these candidates and found that nine of the 26 candidates are highly expressed during avocado colonization (Fig 6A). To further assess *in planta* expression of the nine candidates, we measured their expression in *R. necatrix* strain R18 upon cotton inoculation, revealing that five genes are highly expressed during cotton colonization (Fig 6B). To test whether these *in planta*-expressed predicted secreted proteins indeed possess antimicrobial activity, we pursued heterologous protein production in *E. coli*. Unfortunately, we repeatedly failed to produce four of these proteins (FUN_009266, FUN_0100039, FUN_005751 and FUN_006760) which, although it may be interpreted as a sign of potential antimicrobial activity [67], obstructs functional analysis. However, one protein could be successfully produced and purified (FUN_004580).

FUN_004580 has predicted structural homology to antifungal protein 1 (AFP1) from the bacterium *Streptomyces tendae* [68] (Fig 7A). This protein was identified in a screen for antifungal activity as a non-enzymatic chitin-binding antifungal directed against particular Ascomycete fungi [69]. Intriguingly, attempts to produce functional AFP1 in the lactic acid bacterium *Lactococcus lactis* failed [70], suggesting that the protein may exert antibacterial activity as well. BLAST searches using the FUN_004580 amino acid sequence revealed ~100 Ascomycete fungal homologs, while HMMER searches revealed homologs in the Streptophyta clade (Viridiplantae) too (S6 Table). Interestingly, BLAST and HMMER searches using the amino acid sequence of FUN_004580 did not reveal homology to AFP1, underpinning that

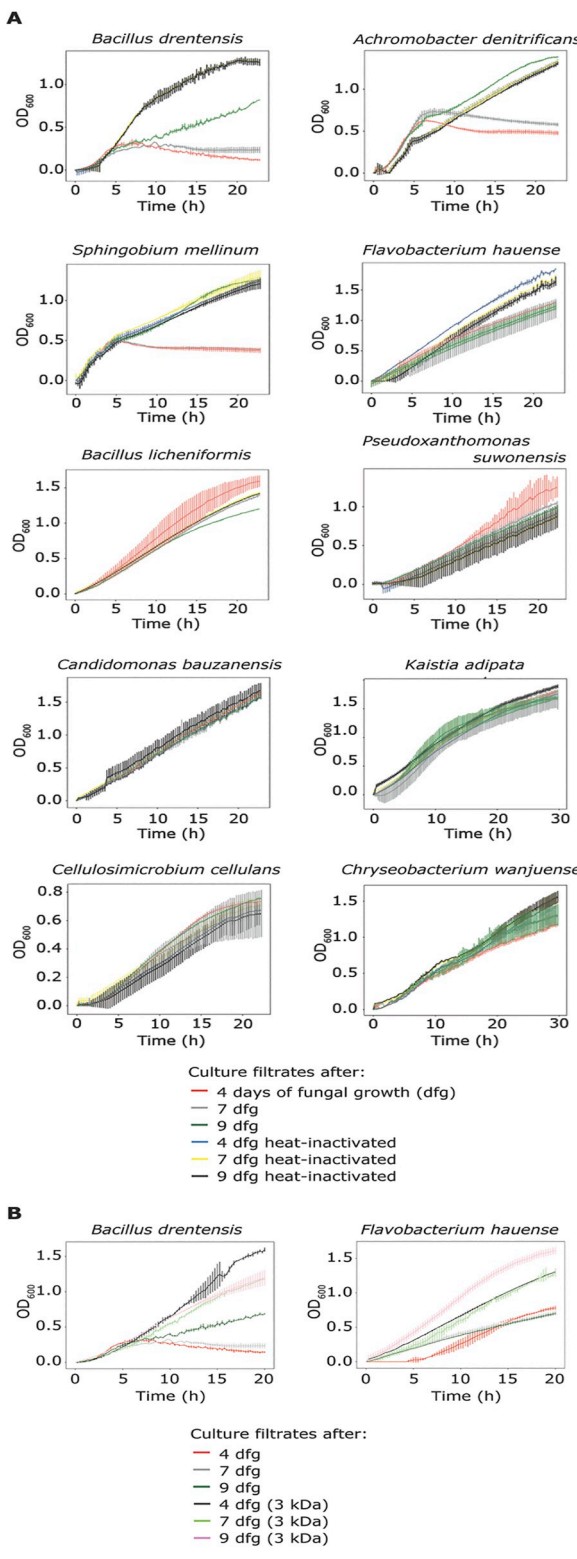

**Fig 5. *Rosellinia necatrix* culture filtrates inhibit the growth of particular plant-associated bacterial species.** (A) Bacterial growth was measured over time in *R. necatrix* culture filtrates, which were obtained after 4 (red), 7 (grey) and 9 (green) days of fungal growth. Heat-inactivated culture filtrates collected after 4 (blue), 7 (yellow) and 9 (black) days of fungal growth were used as controls. (B) Growth of *Bacillus drentensis* and *Flavobacterium hauense* was measured in culture filtrates collected after 4 (red), 7 (grey) and 9 (green) days of fungal growth, and additionally in culture filtrates

passed through spin-columns with 3 kDa cut-off after 4 (black), 7 (light green) and 9 (pink) days of fungal growth. The experiment was performed twice and error bars display standard deviations (n = 3).

structural homology, rather than sequence homology, drove the prediction of FUN_004580 as an antimicrobial.

An alternative manner to obtain proteins for functional analysis that does not rely on expression in heterologous microbial expression systems is peptide synthesis, although this is typically limited to sequences shorter than 100 amino acids. Unfortunately, all five effectors that are expressed in cotton are larger than this size. However, among the nine candidates that

**Table 3. Predicted secreted proteins of *R. necatrix* strain R18 with structural homology to known antimicrobial proteins.**

| *R. necatrix* gene ID | SignalP prediction confidence[a] | Structural homolog[b] | Prediction confidence (%) | Alignment coverage (%)[c] | Homolog protein size (kDa)[d] | Cys (%)[e] | pI[f] |
|---|---|---|---|---|---|---|---|
| FUN_000307 | 96.4 | Bacterial enterotoxin | 38.6 | 14 | 22.2 | 2.8 | 4.86 |
| FUN_000408 | 99.3 | Toxin | 100 | 64 | 40.7 | 3.8 | 5.68 |
| FUN_000625 | 99.3 | Peptidoglycan-binding | 100 | 98 | 12.3 | 4.5 | 8.79 |
| FUN_000907 | 98.4 | Toxin | 40.1 | 35 | 28 | 2.6 | 5.03 |
| FUN_001825 | 99.9 | Hydrolase | 100 | 95 | 38.4 | 1.8 | 5.11 |
| FUN_001865 | 98.7 | EGF/Laminin | 91.4 | 13 | 34.6 | 3.6 | 7.01 |
| FUN_002055 | 99.7 | Endoglucanase | 100 | 89 | 28.6 | 5.9 | 6.73 |
| FUN_003304 | 99.3 | Hydrolase | 100 | 96 | 39.2 | 2.4 | 5.68 |
| FUN_003451 | 99.1 | Fungal ribonucleases | 100 | 92 | 14.7 | 3.6 | 4.19 |
| FUN_004580 | 99.8 | Antifungal protein AFP1 | 55.5 | 6 | 47 | 2.5 | 4.85 |
| FUN_004961 | 97.4 | Toxin | 25.2 | 23 | 19.7 | 3.8 | 7.75 |
| FUN_005431 | 94.8 | Thermolabile hemolysin | 100 | 99 | 32.4 | 0.8 | 4.36 |
| FUN_005483 | 99.6 | NLP | 100 | 87 | 28.9 | 3.3 | 4.87 |
| FUN_005751 | 98.4 | Fungal ribonuclease | 100 | 93 | 14.4 | 3.6 | 6.05 |
| FUN_005940 | 99.4 | Toxin | 40.6 | 17 | 13.9 | 5.8 | 6.63 |
| FUN_006760 | 86.2 | Carboxypeptidase | 100 | 92 | 60.9 | 1.5 | 5.11 |
| FUN_007808 | 91.5 | Toxin | 100 | 55 | 89 | 5.8 | 5.36 |
| FUN_007991 | 98.2 | Toxin | 45.1 | 26 | 19 | 2.6 | 9.00 |
| FUN_008352 | 99.6 | Toxin | 100 | 99 | 16.9 | 6.2 | 6.14 |
| FUN_009115 | 96.1 | Antimicrobial MIAMP1 | 87.8 | 33 | 24 | 3.2 | 4.65 |
| FUN_009151 | 84.6 | Phage lysozyme | 100 | 94 | 20.7 | 3.6 | 6.92 |
| FUN_009266 | 92.6 | Neutrophil defensin 4 | 44.9 | 10 | 15.6 | 3.3 | 7.59 |
| FUN_010039 | 97.8 | Toxin | 100 | 99 | 17.0 | 4.5 | 6.73 |
| FUN_010165 | 99.5 | Toxin | 50.3 | 11 | 17.5 | 5.8 | 4.83 |
| FUN_011519 | 98.6 | Antimicrobial protein | 15.2 | 57 | 5.8 | 5.7 | 4.78 |
| FUN_011592 | 92.8 | Defensin | 49.9 | 10 | 18.8 | 6.7 | 7.41 |

[a]Secreted protein prediction confidence based on SignalP Version 5.0 [44].

[b]Structural homology was determined based on Phyre2 prediction [17].

[c]Alignment coverage of the *R. necatrix* gene query to the database homolog (subject).

[d]Protein size expressed in kilodaltons (kDa) of the database homolog.

[e]Amount of cysteines (Cys) present in the database homolog expressed in percentage (%).

[f]Isoelectric point (pI) of the database homolog.

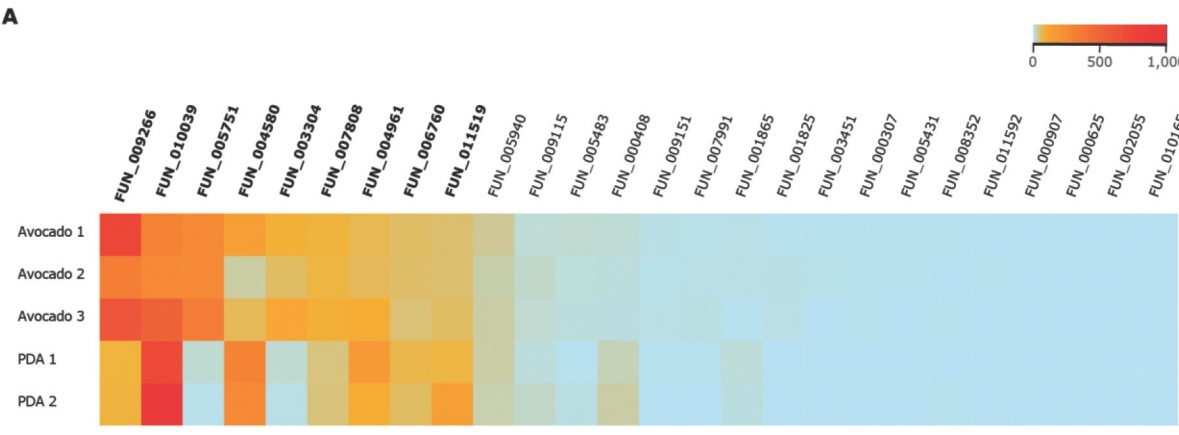

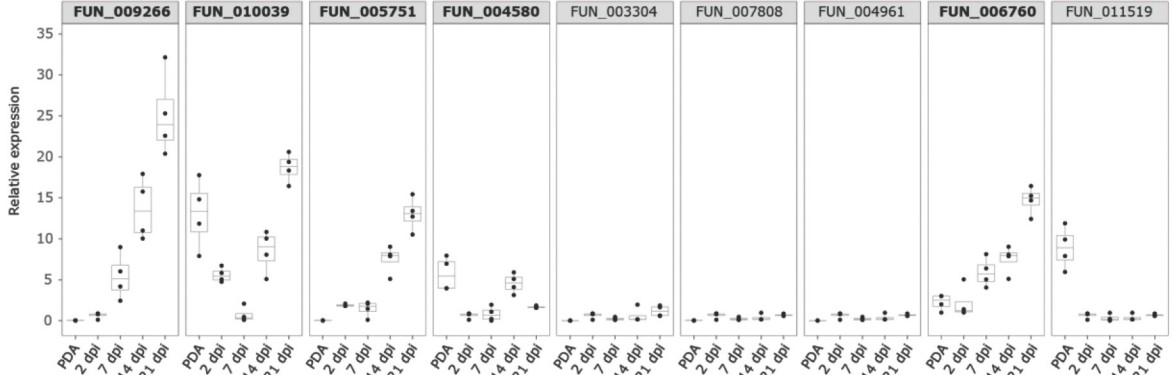

**Fig 6. *In planta* expression of *Rosellinia necatrix* genes predicted to encode effector proteins with antimicrobial activity.** (A) *In silico* analysis of publicly available RNAseq data of avocado plants infected with *R. necatrix* (n = 3) reveals nine genes that are highly induced *in planta* (shown in bold) out of 26 that encode potential antimicrobial proteins (AMPs). Several of these are also expressed upon cultivation of *R. necatrix in vitro* on potato dextrose agar (PDA) (n = 2). (B) Real-time PCR of nine *in planta* expressed *R. necatrix* genes identified in (A) reveals that five genes are expressed during cotton colonization at 14 and 21 days post inoculation (shown in bold).

are highly expressed during avocado colonization, one encodes a protein (FUN_011519) small enough for protein synthesis (Biomatik Corporation, Ontario, Canada).

FUN_011519 has predicted structural homology to the antimicrobial protein AcAMP2 from the plant *Amaranthus caudatus* [71] (Fig 7B). This protein was identified based on homology to cysteine/glycine-rich domain of plant chitin-binding proteins and showed antifungal activity, as well as activity against Gram-positive bacteria [72]. BLAST and HMMER searches revealed only Ascomycete fungal homologs, identifying ~100 fungal proteins (S7 Table). Interestingly, BLAST and HMMER searches using the amino acid sequence of FUN_011519 did not reveal homology to AcAMP2, underpinning that also in this case structural homology, rather than sequence homology, drove the prediction as antimicrobial.

## Candidate antimicrobial effector proteins display selective antimicrobial activity

Given that FUN_004580 and FUN_011519 were identified based on structural homology to proteins with antifungal activity, we tested potential antimicrobial activity on eight fungal

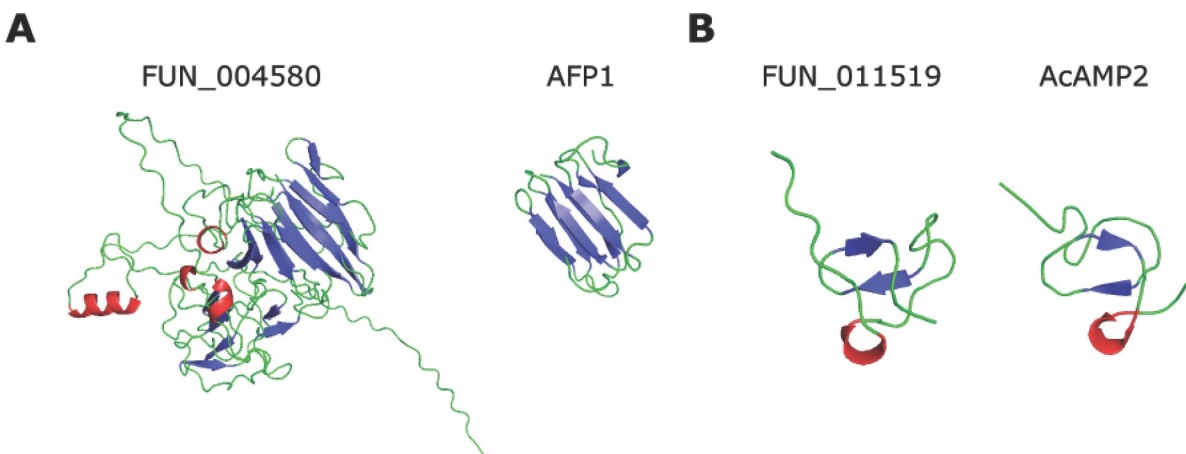

**Fig 7. *Rosellinia necatrix* effector proteins with predicted antimicrobial activity based on structural homology.** (A) Alphafold2 model of *R. necatrix* effector protein FUN_004580 (left) shows structural homology to antifungal protein 1 (AFP1) from the bacterium *Streptomyces tendae* (right). (B) Alphafold2 model of *R. necatrix* effector protein FUN_011519 (left) shows structural homology to antimicrobial protein AcAMP2 from the plant *Amaranthus caudatus* (right).

species: five filamentous fungi (*Verticillium dahliae*, *Alternaria brassicicola*, *Cladosporium cucumerinum*, *Trichoderma viridae* and *R. necatrix*), and four yeasts (*Pichia pastoris*, *Cyberlindnera jadinii*, *Debaryomyces vanrijiae* and *Rhodorotula bogoriensis*). Interestingly, both proteins displayed selective antifungal activity. Whereas FUN_004580 inhibited the growth of all filamentous fungi tested, except *R. necatrix* (Fig 8A) as well as of all yeasts except *P. pastoris* (Fig 7B), FUN_011519 had a narrower activity spectrum as it only inhibited the growth of *V. dahliae* and *A. brassicicola* (Fig 8C), and the yeasts *C. jadinii* and *R. bogoriensis* (Fig 8D). Thus, we conclude that both proteins possess clear antifungal activity.

To further investigate the antimicrobial activity of the two predicted secreted effector proteins FUN_004580 and FUN_011519, we tested their activity on the diversity panel of 37 bacteria that span a wide range of taxonomic diversity, including Gram-positive and Gram-negative taxa (S4 Table). Interestingly, although we found that the growth of most bacteria was not affected by any of the two proteins, growth of some bacteria was clearly hampered. Whereas FUN_004580 strongly impacted growth of *Pseudoxanthomonas suwonensis*, *Flavobacterium hauense*, and *Cellulosimicrobium cellulans*, and inhibited *Chryseobacterium wanjuense* to a lesser extent, FUN_011519 inhibited *Bacillus drentensis* and *Achromobacter denitrificans*, and possibly also *Sphingobium mellinum* and *Candidimonas bauzanensis* (Fig 9). Thus, we conclude that both effectors display antibacterial activity as well. Collectively, our data show that both effector proteins possess selective antimicrobial activity.

## Several plant-associated bacteria display antagonistic activity

One obvious reason for a fungus to target particular microbes with antimicrobials is that these may be detrimental to fungal growth due to the display of antagonistic activities. To test for such activities directed against *R. necatrix*, we conducted confrontation assays *in vitro*, where we grew *R. necatrix* near the individual bacteria species from the diversity panel of 37 bacteria (S4 Table). In these assays, we observed antagonistic effects of several bacterial species on *R. necatrix* (Fig 10). Interestingly, several of the bacterial species inhibited by either of the two effector proteins displayed antagonistic activity against *R. necatrix*, including *Bacillus drentensis*, *Bacillus licheniformis*, and *Sphingobium mellinum* (inhibited by FUN_011519), and *Cellulosimicrobium cellulans*, *Chryseobacterium wanjuense* and *Pseudoxanthomonas suwonensis*

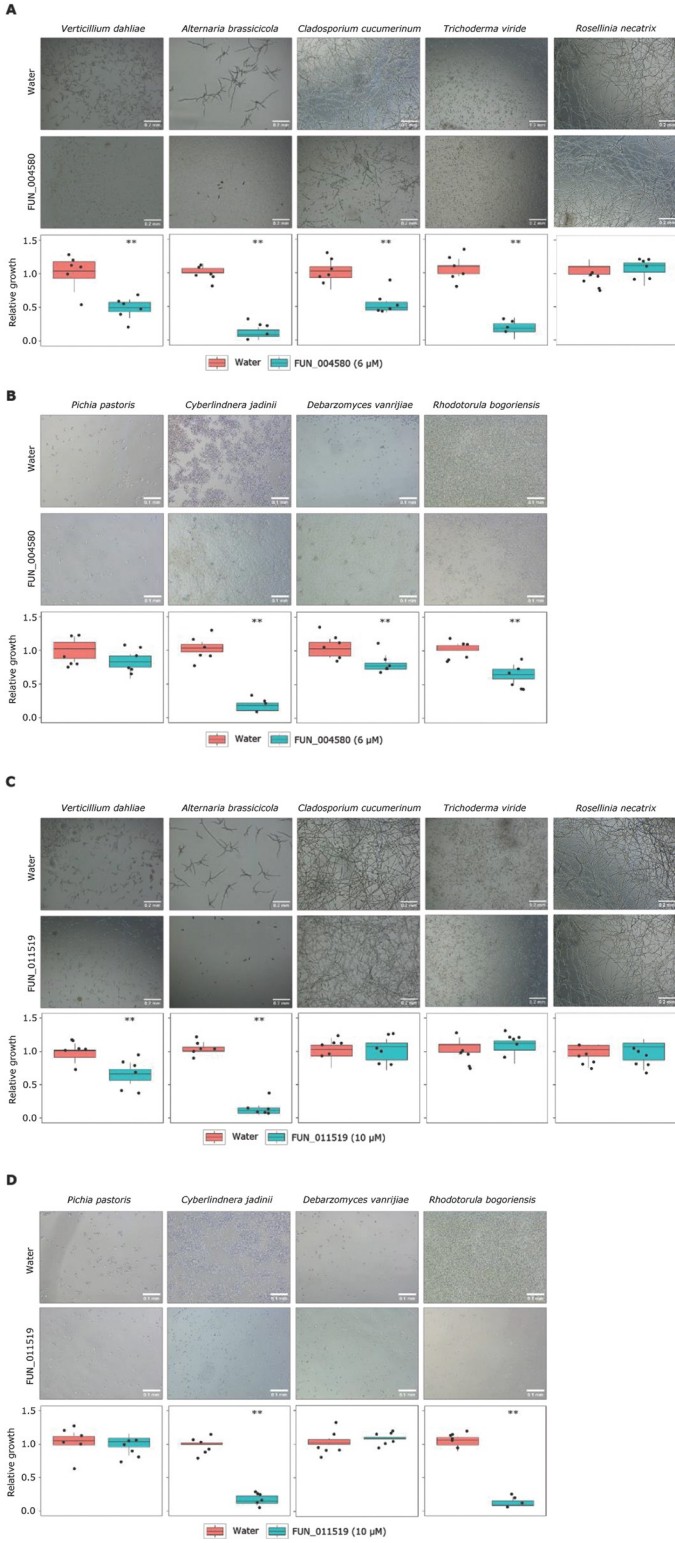

**Fig 8. *Rosellinia necatrix* effector proteins display antifungal activity.** Fungal growth in 5% potato dextrose broth (PDB) in presence of FUN_004580 (6 μM) or water of filamentous fungal species (A) and yeasts (B), or in presence of FUN_011519 (10 μM) or water of filamentous fungal species (C) and yeasts (D). Relative growth as display in microscopic pictures (n = 6) was quantified using ImageJ. Asterisks indicate significant differences (unpaired two-sided Student's t test (P<0.01)).

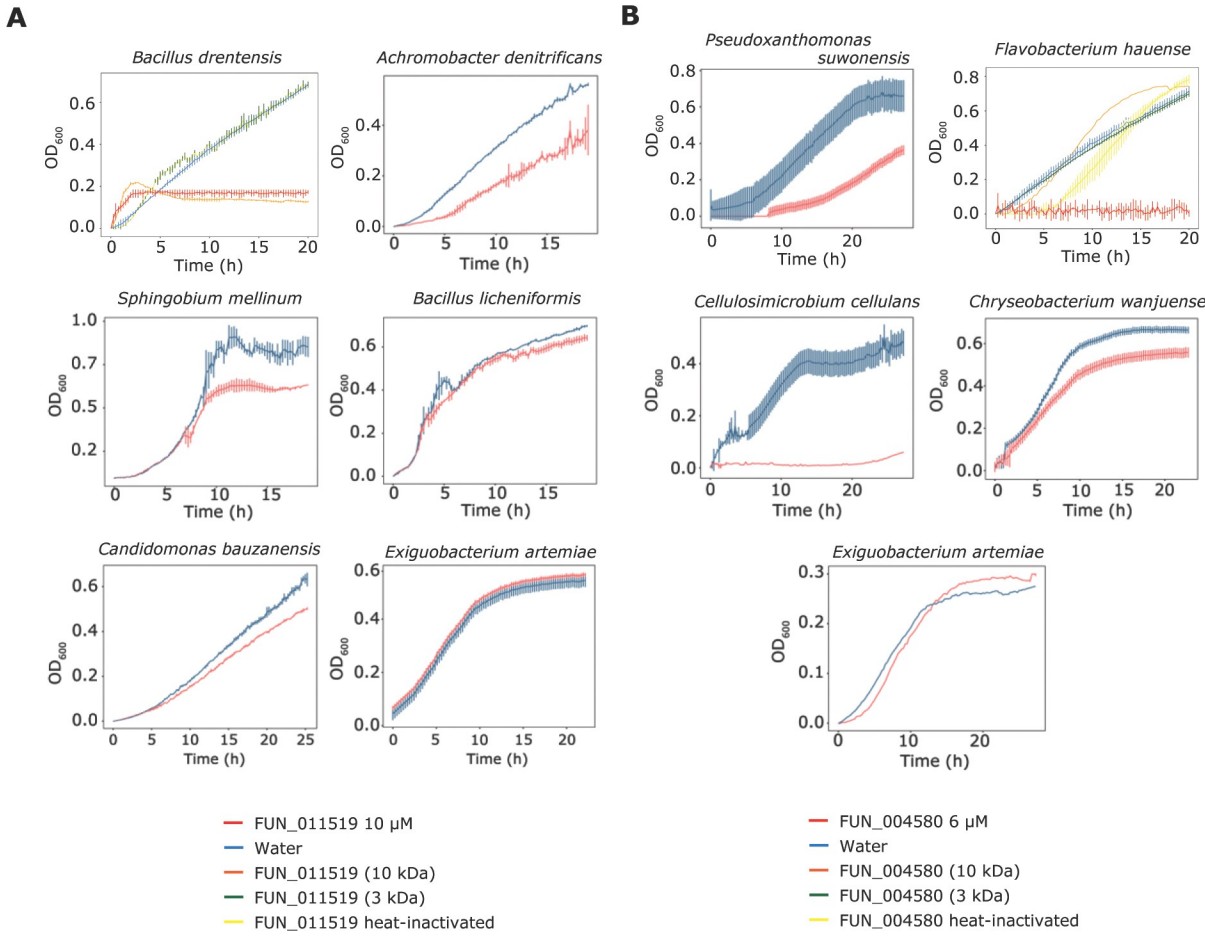

**Fig 9. *Rosellinia necatrix* effector proteins display selective antimicrobial activity against plant-associated bacteria.** (A) Red lines show bacterial growth in presence of FUN_011519 (10 µM) while blue lines show bacterial growth in water. For *Bacillus drentensis* additional orange line shows bacterial growth in protein filtered with spin-column with 10 kDa cut off, green line shows bacterial growth in protein filtered with spin-column with 3 kDa cut off, and yellow line shows growth in heat-inactivated protein. (B) Red lines show bacterial growth in presence of FUN_004580 (6 µM) while blue lines show bacterial growth in water. For *Flavobacterium hauense* additional orange line shows bacterial growth in protein filtered with spin-column with 10 kDa cut off, green line shows bacterial growth in protein filtered with spin-column with 3 kDa cut off, and yellow line shows growth in heat-inactivated protein. Error bars display standard deviations (n = 3). Growth of *Exiguobacterium artemiae* is not inhibited by either protein and is shown as a representative of non-inhibited bacterial taxa.

(inhibited by protein FUN_004580) (Fig 10). However, also some other bacterial species that were not found to be inhibited by either of the two effector proteins displayed antagonistic activity, namely *Kaistia adipata*, *Pedobacter steynii*, *Pedobacter panaciterrae*, *Pseudomonas corrugata*, *Pseudomonas knackmusii*, *Solibacillus isronensis* and *Solibacillus silvestris*. These findings suggest that a diversity of plant-associated bacteria possess inherent antagonistic activity against *R. necatrix* and that some of these bacteria are targeted by effector proteins of *R. necatrix*.

## Particular antagonists alleviate Rosellinia disease in cotton

To test whether any of the bacterial species that are inhibited by the *R. necatrix* effector proteins and that display antagonistic activity against the fungus can inhibit disease development, we pursued inoculation assays in the presence and absence of these bacteria. As *R. necatrix* is a broad host range pathogen, we performed infection assays on cotton plants. To this end,

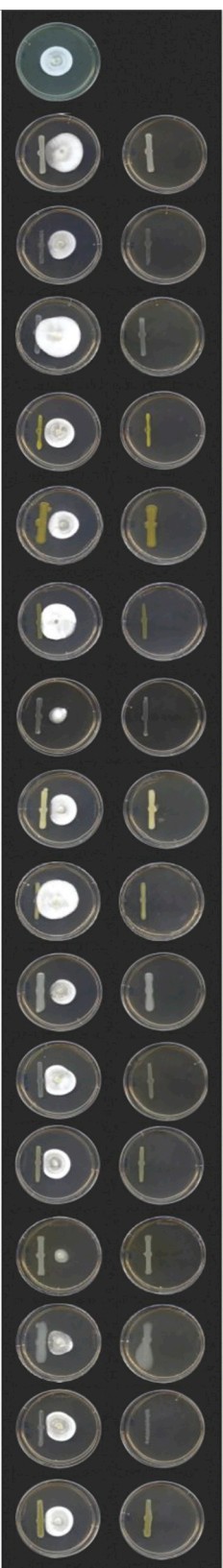

*Rosellinia necatrix* R18

*Bacillus drentensis*

*Bacillus licheniformis*

*Achromobacter denitrificans*

*Sphingobium mellinum*

*Chryseobacterium wanjuense*

*Flavobacterium hauense*

*Pseudoxanthomonas suwonensis*

*Cellulosimicrobium cellulans*

*Candidimonas bauzanensis*

*Kaistia adipata*

*Pedobacter steynii*

*Pedobacter panaciterrae*

*Pseudomonas corrugata*

*Pseudomonas knackmusii*

*Solibacillus isronensis*

*Solibacillus silvestris*

**Fig 10. Various plant-associated bacteria display antagonistic activity against R***osellinia necatrix***.** Fungal PDA disks were placed in the center of a petri dish and next to it individual bacteria species were deposited in a straight line with help of a spreader (n = 5), bacteria species were placed alone (right column) as controls.

14-day-old cotton cv. XLZ seedlings were inoculated with *R. necatrix* strain R18, leading to clear disease symptoms by two weeks after inoculation (Fig 11). Next, we pre-treated cotton cv. XLZ seeds with antagonistic bacteria *Bacillus drentensis*, *Sphingobium mellinum*, *Cellulosimicrobium cellulans*, or *Pseudoxanthomonas suwonensis*. To show that the inhibition of disease development by the antagonistic bacteria is not caused by the activation of MAMP-triggered host immunity we included pre-treatment with the non-antagonistic bacteria *Exiguobacterium artemiae*, *Candidomonas bauzanensis*, *Flavobacterium hauense*, *Achromobacter denitrificans*, or soil. Furthermore, a mix of all bacteria including the antagonistic and non-antagonistic bacteria species was included as well as water as a control, followed by *R. necatrix* inoculation. Interestingly, two weeks after inoculation it could be observed that pretreatment with each of the antagonistic bacteria resulted in reduced susceptibility of the cotton plants to *R. necatrix* when compared with the plants that did not receive a bacterial pre-treatment. Because treatment with the non-antagonistic bacterial species or soil suspension did not lead to reduced disease symptoms, we infer that the reduced disease development upon treatment with then antagonistic bacteria is unlikely to be caused by the activation of MAMP-triggered host immunity, but rather likely through direct antagonism towards the fungal pathogen. Finally, none of the bacteria affected growth of the cotton plants in absence of pathogen inoculation (Fig 11).

## Discussion

Increasing evidence supports the notion that plants can recruit microbes into their microbiota that can help them to withstand pathogen infection [73–76]. Based on this notion, it has been hypothesized that pathogens have evolved to counter-act the recruitment of beneficial microbes to promote host colonization [77]. In support of this hypothesis, it has been shown that the soil-borne fungal pathogen *Verticillium dahliae* secretes a suite of effector molecules with selective antimicrobial activities to suppress antagonistic microbes in host microbiota to support host ingress [19–21]. Although it has been hypothesized that other fungal pathogens are likely to have evolved to modulate host microbiota as well [19,21], evidence for this hypothesis has been lacking thus far. In this study, we show that the soil-borne pathogen *Rosellinia necatrix* similarly expresses predicted secreted effector proteins during host colonization with selective antimicrobial activity. Specifically, we show that two genes encoding the effector proteins, FUN_004580 and FUN_011519, are expressed during host colonization and these predicted secreted effectors display selective antimicrobial activity *in vitro* towards specific plant-associated bacteria and fungi. Intriguingly, some of the bacterial species that are inhibited by the antimicrobial effector proteins display antagonistic activity towards *R. necatrix in vitro*. Moreover, when applied to cotton seeds, several of the antagonistic bacterial species reduce white root rot disease. Collectively, our findings suggest that *R. necatrix* secretes antimicrobial effector proteins during host colonization to modulate host microbiota and support disease development.

The effector proteins FUN_004580 and FUN_011519 were found based on structural homology to AFP1 from the bacterium *Streptomyces tendae* and homology to AcAMP2 from the plant *Amaranthus caudatus*, respectively. AFP1 has been described as a non-enzymatic chitin-binding antifungal directed against particular Ascomycete fungi [69], and recent studies have shown that a homolog of AFP1 isolated from the plant *Carum carvi* (Cc-AFP1) alters fungal cell membrane permeability [78]. AcAMP2 was identified based on homology to cysteine/

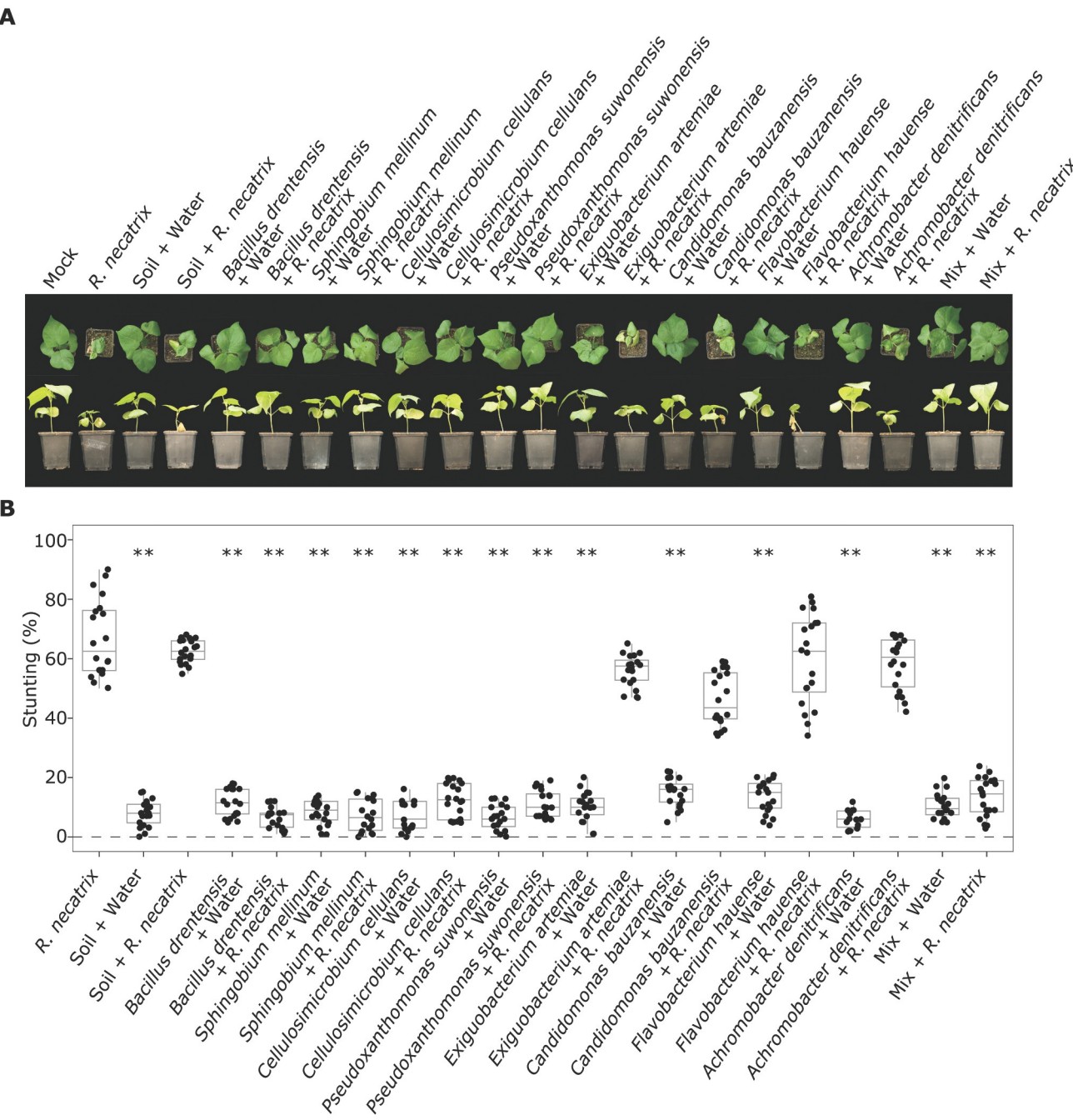

**Fig 11. Antagonistic plant-associated bacteria reduce white root rot disease in cotton.** (A) Top and side pictures of cotton plants. Cotton seeds were soaked in a suspension of the antagonistic bacteria *Bacillus drentensis*, *Sphingobium mellinum*, *Cellulosimicrobium cellulans*, or *Pseudoxanthomonas suwonensis*, and the non-antagonistic bacteria *Exiguobacterium artemiae*, *Candidomonas bauzanensis*, *Flavobacterium hauense*, or *Achromobacter denitrificans*, a mix of all bacteria, soil suspension, or water as a control. Subsequently, germlings (n = 10) were inoculated with *R. necatrix* strain R18 or mock-inoculated. (B) Quantification of stunting. Asterisks indicate significant differences between *R. necatrix*-inoculated plants and plant seeds treated with bacteria (ANOVA followed by Fisher's LSD test (P<0.01)). The experiment was performed twice.

glycine-rich domain of plant chitin-binding proteins and showed antifungal activity, as well as activity against Gram-positive bacteria [72]. The antimicrobial mechanism described for the structural homologs suggest that FUN_004580 and FUN_011519 could potentially be involved

in cell membrane disruption, but further functional studies that also include antifungal activity assays are required to investigate this.

The effector proteins FUN_004580 and FUN_011519 showed selective antimicrobial activity against plant-associated bacterial and fungal species *in vitro* and display divergent activity spectra, suggesting that they possess additive activities. Similar observations have been made for the antimicrobial effector proteins VdAMP2 and VdAve1 of *V. dahliae* that add to each other's activity spectra during soil colonization [19]. Divergent activity spectra have been also observed in the *in planta*-expressed effectors VdAve1, VdAve1L2 and VdAMP3 [19,21]. Furthermore, differential expression of antimicrobial effectors on different plant hosts was observed in *R. necatrix*. Out of the 26 AMP candidates, nine are expressed during avocado colonization, and five of them were also expressed during cotton colonization. From these, *FUN_004580* and *FUN_011519* are expressed during avocado colonization, while only *FUN_004580* is expressed on cotton (Fig 4). The expression profiles on different host plants suggest differential expression of antimicrobials in *R. necatrix*, potentially due to differential microbiota compositions of these plant hosts. Possibly, FUN_011519 evolved to target microbes that are specifically associated with avocado, or absent from the microbiota of cotton plants.

Various studies have shown that plants shape their microbiota composition [79]. For example, one single gene responsible for the production of glucosinolate significantly altered the microbial community on the roots of transgenic *Arabidopsis* [80]. Moreover, the ATP-binding cassette (ABC) transporter *abcg30* mutant of *Arabidopsis thaliana* secretes root exudates that are enriched in phenolic compounds and contain fewer sugars, resulting in distinct root microbiota compositions [81]. Furthermore, *A. thaliana* genes responsible for defense, kinase-related activities, and cell wall integrity impact the microbiota composition [82]. Besides *A. thaliana*, different potato and tomato plants assemble different microbiota compositions [83,84].

Like *V. dahliae*, *R. necatrix* is a soil-borne pathogen that resides for part of its life cycle in the microbe-rich soil where inter-microbial competition for limited nutrients is ferocious. Although *R. necatrix* is phylogenetically related to *V. dahliae*, both belonging to the class of Sordariomycetes, our findings suggest that exploitation of antimicrobial effector proteins among soil-borne fungi may be commonplace. However, whether the employment of such effector proteins is relevant for soil-borne pathogens only, or shared by other types of plant pathogens too, presently remains enigmatic [21].

To further investigate the effector catalog of *R. necatrix*, we investigated whether effector genes in *R. necatrix* are associated with repetitive regions. The genomes of many filamentous plant pathogens are compartmentalized, comprising gene-dense/repeat-poor regions harboring essential and widely conserved housekeeping genes and gene-sparse/repeat-rich regions containing fast-evolving virulence-associated genes [55,85,86]. However, in the case of *R. necatrix*, the repeat content is low, and interestingly, the vast majority of repetitive elements are not located in close proximity to effectors. This suggests that effector genes in *R. necatrix* are not associated with repeat-rich regions. Similar to *R. necatrix*, other genomes have been found to lack obvious genome compartmentalization such as that of the leaf spot fungus *Ramularia collo-cygni* [87], the barley powdery mildew fungus *Blumeria graminis* f.sp. *hordei* [88] and the wheat stripe rust fungus *Puccinia striiformis* f.sp. *tritici* [89]. Intriguingly, while the genome of the latter two is characterized by a high repeat content, the genome of *Ramularia collo-cygni* has a low repeat content like *R. necatrix* [87–89]. In such genomes, adaptation and evolution of effector genes were suggested to be governed by copy-number variation (CNV) and heterozygosity of the effector loci [88]. In *R. necatrix*, we showed that insertions and deletions are predominant structural variants (SVs) that are associated with effector genes, which suggests that these SVs may play a role in evolution of the effector catalog.

To understand the diversity and functional properties of effectors, we investigated their 3D protein structure. The results of this study demonstrate that there is relatively little structural clustering among *R. necatrix* effectors. The five largest clusters identified were mainly driven by sequence conservation rather than structural similarity, suggesting that effector family expansion based on structural similarity played only a minor role in *R. necatrix* effector evolution. Effector protein structure clustering has revealed that many plant pathogen effectors fall into distinct structural families. For example, MAX [65], and RNase-Like proteins associated with haustoria (RALPH) [90] are effector families with structural similarity. Common themes are emerging like structural homology and in some cases similar modes of action [91]. While structural conservation has been observed in some pathogenic fungi [65,90], sequence conservation is the main driver of structural clustering among *R. necatrix* effectors. Future studies could explore the functional significance of the identified effector clusters in *R. necatrix* and other plant pathogens and investigate the mechanisms underlying the evolution of effector proteins.

## Materials and methods

### Nanopore sequencing and genome assembly

High-molecular weight (HMW) DNA was extracted from nine *R. necatrix* strains (Table 1) as described by Chavarro-Carrero et al. (2021) [92], except that mycelium was grown in 1/5 PDB instead of Czapek Dox medium. DNA quality, size and quantity were assessed with Nanodrop, Qubit analyses, and with gel electrophoresis. Library preparation with the Rapid Sequencing Kit (SQK-RAD004) was performed with ~400 ng HMW DNA according to the manufacturer's instructions (Oxford Nanopore Technologies, Oxford, UK). An R9.4.1 flow cell (Oxford Nanopore Technologies, Oxford, UK) was loaded and ran for 24 h, and subsequent base calling was performed using Guppy (version 3.1.3; Oxford Nanopore Technologies, Oxford, UK). Adapter sequences were removed using Porechop (version 0.2.4 with default settings; [93] and the reads were self-corrected, trimmed and assembled using Canu (version 1.8; [94].

To obtain longer reads ultra-high-molecular weight (UHMW) DNA extraction was performed. To this end, after DNA precipitation, the Nanobind Big DNA Kit (SKU NB-900-801-01, Circulomics, USA) was used following the manufacturer's protocol to select DNA fragments >50 kb. Simultaneously, to allow error correction of the ONT reads, paired-end short read (150 bp) sequencing was performed at the Beijing Genome Institute (BGI) using the DNBseq platform. After error correction using FMLRC (version 1.0.0 with default settings; [95], ONT reads were trimmed and assembled using Canu.

Hi-C was performed based on the Proximo Hi-C kit (Microbe) (Phase Genomics, Seattle, WA, USA) according to the manufacturer's protocol as previously described [96], and the Hi-C library was paired-end (150 bp) sequenced on the NextSeq500 platform at USEQ (Utrecht, the Netherlands). Subsequently, Hi-C reads were mapped to the Canu assembly using Juicer (version 1.6 with early-stage setting; [97], and the three-dimensional (3D) *de novo* assembly (3D-DNA) pipeline (version 180922 with contig size threshold 1000; [98] was used. The genome assembly was further manually improved using Juicebox Assembly Tools (JBAT) (version 1.11.08; [99] and the 3D-DNA post review asm pipeline [98]. Tapestry (version 1.0.0 with default settings; [26] was used to visualize chromosomes and telomeric regions with telomeric sequence ([TTAGGG]*n*).

### Phylogenetic tree construction and structural variant calling

A phylogenetic tree of all sequenced strains was constructed with Realphy (version 1.12) [61]. Briefly, genomic reads were mapped against the genome of strain R18 using Bowtie2 (version

2.2.6, [64]) and a maximum-likelihood phylogenetic tree was inferred using RAxML (version 8.2.8) [100], using the genome sequence of *Rosellinia corticium* to root the tree. Subsequently, to predict SVs, we used NanoSV (version 1.2.4 with default settings; [62]) for each of the genomes when compared to R18 strain as reference. The output of each strain was initially filtered with bcftools v.1.3.2 using GT = AA, QUAL>50, GQ>10, SUPPORT>20 [101]. The results were merged using SURVIVOR v.1.0.6 [102], allowing 1,000 bp as the maximum distance for breakpoints and considering only same SV types. Only SVs with minimum size >100 bp and maximum size of 100 kb were kept. Permutation tests were computed using R/ Bioconductor region v.1.8.1 package [103]., and 10,000 iterations were performed using the mean distance to evaluate the closest relationship (bp distance) between SV-breakpoints, TEs and effectors, and circular randomization as well to maintain the order and distance of the regions on the chromosomes.

## Genome mining

Genome annotation was performed using Funannotate (Version 1.8.13, [28]) guided by publicly available RNAseq data [8]. Furthermore, secreted genes, effector genes and secondary metabolite gene clusters were predicted using SignalP (Version 5.0, [44]), EffectorP (Version 2.0, [29]) and antiSMASH (Version 6.0.0, [45]), respectively. The *R. necatrix* R18 genome was mined for gene candidates that encode secreted antimicrobials as described previously [19]. Briefly the predicted secretome and effector catalog were mined using Phyre2 [17], followed by manual curation, to search for predicted structural homologues of known antimicrobial proteins. Gene sequences were extracted using Bedtools (setting: getfasta) [104]. Subsequently, RNAseq data were used to confirm expression of selected candidates. To this end, RNAseq data was mapped to the R18 assembly and relative expression was calculated using Kallisto (version 0.46.1, default settings, [105]).

Repetitive regions were annotated using EDTA v.1.9.4 [106]. Briefly, we used the 'sensitive', 'anno' and 'evaluate' options to maximize the transposable element discovery and integrity. Then, the TE library was refined using one code to find them all [107], and only full LTRs were kept.

## Structural prediction of the *R. necatrix* effector catalog

Structural predictions of effector candidates were done based on mature amino acid sequences, lacking signal peptides, with Alphafold2 [108] using the casp14 preset. Five models were generated for each effector candidate, of which the one with the highest pLDDT score was used for further analysis. To assess structural similarity between effector candidates, an all-vs-all alignment was performed using TM-align [109]. Structures were considered similar when the average TM-score of pairwise reciprocal alignments, calculated on a 0–1 scale, was >0.5. Effector clusters were identified by simple agglomerative hierarchical clustering based on structural similarity using SciPy [110] and a minimum similarity threshold of 0.5. For the five largest clusters the average sequence identity was calculated from a multiple sequence alignment performed with MAFFT on EMBL-EBI [111]. Cluster annotation was performed by generating PANNZER [112] annotation for each protein followed by manual curation of a consensus annotation for each cluster. Protein structures were generated with PyMOL (The PyMOL Molecular Graphics System, Version 2.0 Schrödinger, LLC, DeLano Scientific, Palo Alto, CA, USA).

## Real-time PCR

To determine expression profiles of candidate genes that encode secreted antimicrobials during *R. necatrix* infection of cotton, two-week-old cotton (cv. XLZ) seedlings were inoculated,

as previously described [113], with *R. necatrix* strain R18, and stems were harvested up to 21 dpi. Furthermore, mycelium was harvested from ten-day-old *R. necatrix* strain R18 cultures on potato dextrose agar (PDA) plates. Total RNA extraction and cDNA synthesis were performed as previously described [114]. Real time-PCR was performed with primers listed in S5 Table, using the *R. necatrix* glyceraldehyde-3-phosphate dehydrogenase gene (GAPDH; FUN_007054) as endogenous control. The PCR cycling conditions consisted of an initial 95˚C denaturation step for 10 min followed by denaturation for 15 s at 95˚C, annealing for 30 s at 60˚C, and extension at 72˚C for 40 cycles.

## Heterologous protein production

The sequences encoding mature proteins were cloned into vector pET-15b with an N-terminal His6 tag sequence (Novagen, Darmstadt, Germany) (primer sequences, see S5 Table). The resulting expression vectors were verified by sequencing and used to transform *Escherichia coli* strain BL21. For heterologous protein production, BL21 cells were grown in 2× yeast extract tryptone (YT) medium at 37˚C with continuous shaking at 200 rpm until the cultures reached an optical density measurement at 600 nm ($OD_{600}$) of 2. Protein production was induced with 1 mM isopropyl β-D-1-thiogalactopyranoside (IPTG) at 42˚C and shaking at 200 rpm for 2 h. Bacterial cells were pelleted and washed with 50mM Tris HCl and 10% glycerol at pH 8.0. To disrupt cells, they were sonicated three times according to a cycle composed of one second sonication, followed by 0.5 seconds on ice and a one-minute pause. Next, the samples were centrifugated at 16,000g for 10 min. The insoluble pellets were resuspended in 30 mL of 6 M guanidine hydrochloride (GnHCl), 10 mM Tris at pH 8.0 and 10 mM β-mercaptoethanol and incubated overnight at room temperature. Next, debris was pelleted by centrifuging at 16,000g for 10 min and the resulting protein solution was transferred to a new tube and filtered stepwise from 1.2 to 0.45 μm. Proteins were subsequently purified under denaturing conditions by metal affinity chromatography using a pre-packed HisTrap FF column (Cytiva, Medemblik, The Netherlands) and dialysed (Spectra/Por 3 Dialysis Membrane, molecular weight cut off of 10 kDa; Spectrum Laboratories, Rancho Dominguez, U.S.A) against 20 volumes of 0.01 M Bis-Tris, 10 mM reduced glutathione and 2 mM oxidized glutathione at pH 8.0 with decreasing GnHCl concentrations in five consecutive steps of minimum 24 h from 5 M to 1 M for refolding. Finally, proteins were dialysed against demineralized water for 24 h and final concentrations were determined using Qubit (ThermoFisher Scientific, Waltham, U.S.A).

## Culture filtrate production

*R. necatrix* strain R18 was grown in potato dextrose broth (PDB) at 21˚C and 130 rpm. Cultures were collected at 4, 7 and 9 days after the start of incubation and mycelium was separated from the culture medium using a 0.45 nm filter. Half of the culture-filtrate was heat-inactivated at 96˚C for 10 minutes. Finally, culture filtrates were stored at -20˚C until use.

## *In vitro* antimicrobial activity assays

Growth assays were performed as previously described [19]. Briefly, bacterial strains were grown overnight on lysogeny broth agar (LBA), tryptone soya agar (TSA) or R2A agar at 28˚C (S4 Table). Subsequently, single colonies were selected and grown overnight at 28˚C while shaking at 130 rpm in low-salt Luria broth (LB) (NaCl, 0.5 g/L; Tryptone, 10 g/L; Yeast Extract, 5 g/L). Overnight cultures were resuspended to $OD_{600} = 0.05$ in low-salt LB and 100 μL were supplemented with 100 μL of purified effector proteins or culture filtrates to a final $OD_{600} = 0.025$ in clear 96 well flat-bottom polystyrene tissue culture plates (Greiner CELLSTAR, Sigma, Burlington, U.S.A). Culture plates were incubated in a CLARIOstar plate reader (BMG

LABTECH, Ortenberg, Germany) at 25°C with double orbital shaking every 15 minutes and $OD_{600}$ was measured every 15 minutes.

Filamentous fungal isolates were grown on PDA at 22°C, while for yeasts, single colonies were selected and grown overnight in 5% PDB at 28°C while shaking at 200 rpm. For yeasts overnight cultures were resuspended to $OD_{600}$ = 0.01 in fresh 5% PDB supplemented with water, 10 μM FUN_011519 or 6 μM FUN_004580. For filamentous fungi, spores were harvested from PDA and suspended in 5% PDB supplemented with 10 μM FUN_011519, 6 μM FUN_004580 or water to a final concentration of $10^4$ spores/mL. Next, 200 μL of the fungal suspensions was aliquoted in clear 96-well flat bottom polystyrene tissue-culture plates (Greiner CELLSTAR, Sigma, Burlington, U.S.A). Plates were incubated at 28°C, and fungal growth was imaged using an SZX10 stereo microscope (Olympus, Tokyo, Japan) with EP50 camera (Olympus, Tokyo, Japan).

## Microbial confrontation assays

*R. necatrix* was grown on PDA for 14 days and mycelium disks of 5 mm diameter were taken using a biopsy puncher (Kai Europe, Solingen, Germany). Single *R. necatrix* disks were placed in the center of a PDA plate and 100 μL ($OD_{600}$ = 0.05) of bacterial culture (S4 Table), grown overnight in LB medium at 28°C at 130 rpm, was added in proximity of the fungal disk with a spreader (n = 5). After seven days, pictures were taken. Experiment was repeated three times.

## Biocontrol assay

*Bacillus drentensis*, *Sphingobium mellinum*, *Cellulosimicrobium cellulans* and *Pseudoxanthomonas suwonensis* were grown overnight at 28°C in low-salt LB while shaking at 130 rpm. Subsequently, the bacterial suspension was pelleted, washed, and resuspended to $OD_{600}$ = 0.05 in water. Seeds of cotton cv. XLZ were surface-sterilized with 2% sodium hypochlorite for ten minutes, washed three times with sterile water, and then soaked for ten minutes the bacterial suspension. Next, the seeds were transferred to sterile Petri dishes containing filter paper moistened with each of the bacterial suspension and incubated in darkness at room temperature. Once the seeds germinated, germlings were transferred to potting soil in the greenhouse (16 hours light at 24°C and 8 hours darkness at 22°C). Seven days after transfer to potting soil, the seedlings were inoculated with mycelium of a 14-day-old *R. necatrix* culture in PDB. To this end, the mycelium was pelleted, washed, and resuspended in 250 mL water. This mycelial suspension was subsequently used for root-dip inoculation of the cotton seedlings for 10 min, after which they were re-planted and 10 mL of mycelium suspension was poured onto the pot in close proximity to the stem. At 14 days post inoculation (dpi) disease symptoms were recorded. Experiment was repeated three times.

## Supporting information

**S1 Table. Annotation of predicted effector proteins of *R. necatrix* strain R18.**
(DOCX)

**S2 Table. Single nucleotide polymorphism ratios (SNPs in %) and presence-absence variation for predicted effector genes in the *R. necatrix* strains sequenced in this study using *R. necatrix* strain R18 as a reference.**
(DOCX)

**S3 Table. Annotated secondary metabolite clusters of *R. necatrix* strain R18.**
(DOCX)

**S4 Table. Bacterial strains used in this study.**
(DOCX)

**S5 Table. Primers used in this study.**
(DOCX)

**S6 Table. Annotation of BLAST and HMMER hits to effector FUN_004580.**
(DOCX)

**S7 Table. Annotation of BLAST and HMMER hits to effector FUN_011519.**
(DOCX)

## Author Contributions

**Conceptualization:** Edgar A. Chavarro-Carrero, Nick C. Snelders, Michael F. Seidl, Bart P. H. J. Thomma.

**Formal analysis:** Edgar A. Chavarro-Carrero, Nick C. Snelders, David E. Torres, Anton Kraege.

**Funding acquisition:** Edgar A. Chavarro-Carrero, David E. Torres, Ana López-Moral, Bart P. H. J. Thomma.

**Investigation:** Edgar A. Chavarro-Carrero, Nick C. Snelders, David E. Torres, Anton Kraege.

**Resources:** Ana López-Moral, Gabriella C. Petti, Wilko Punt, Jan Wieneke, Rómulo García-Velasco, Carlos J. López-Herrera.

**Supervision:** Nick C. Snelders, Michael F. Seidl, Bart P. H. J. Thomma.

**Writing – original draft:** Edgar A. Chavarro-Carrero, Nick C. Snelders, David E. Torres, Anton Kraege.

**Writing – review & editing:** Michael F. Seidl, Bart P. H. J. Thomma.

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
