## [Decision Letter · Decision Letter 0]

12 Jun 2023

Dear Dr. Thomma,

Thank you very much for submitting your manuscript "The soil-borne white root rot pathogen  Rosellinia necatrix  expresses antimicrobial proteins during host colonization" for consideration at PLOS Pathogens. As with all papers reviewed by the journal, your manuscript was reviewed by members of the editorial board and by several independent reviewers. In light of the reviews (below this email), we would like to invite the resubmission of a significantly-revised version that takes into account the reviewers' comments.

Your work has been assessed by three reviewers who all consider your work of relevance and importance as a case study for molecular warfare between fungal pathogens and plant microbiota. However, two reviewers raised major issues. Please address the major points raised by reviewers 1 and 2 in a revised version. This will require additional, but mostly straightforward experimentation, and a major revision of the manuscript text.

In addition to the suggested experiments by reviewer 1 and if experimentally possible provide a phenotypic assessment of loss-of-function mutants in R. necatrix for FUN_004580 and FUN_011519 or a technical or scientific justification on why these mutants had not been generated and tested.

In the revised manuscript, please make sure to emphasise that predicted secreted proteins are in the focus of this study or provide experimental evidence for their secretion.

We cannot make any decision about publication until we have seen the revised manuscript and your response to the reviewers' comments. Your revised manuscript is also likely to be sent to reviewers for further evaluation.

Sincerely,

Sebastian Schornack, Ph.D.

Academic Editor

PLOS Pathogens

Shou-Wei Ding

Section Editor

PLOS Pathogens

Kasturi Haldar

Editor-in-Chief

PLOS Pathogens

orcid.org/0000-0001-5065-158X

Michael Malim

Editor-in-Chief

PLOS Pathogens

orcid.org/0000-0002-7699-2064

Your work has been assessed by three reviewers who all consider your work of relevance and importance as a case study for molecular warfare between fungal pathogens and plant microbiota.

However, two reviewers raised major issues. Please address the major points raised by reviewers 1 and 2 in a revised version.

This will require additional, but mostly straightforward experimentation, and a major revision of the manuscript text.

In addition to the suggested experiments by reviewer 1 and if experimentally possible provide a phenotypic assessment of loss-of-function mutants in R. necatrix for FUN_004580 and FUN_011519 or a technical or scientific justification on why these mutants had not been generated and tested.

In the revised manuscript, please make sure to emphasise that predicted secreted proteins are in the focus of this study or provide experimental evidence for their secretion.

Reviewer's Responses to Questions

**Part I - Summary**

Reviewer #1: The manuscript by Chavarro-Carrero and co-workers reports the establishment of a gapless genomic assembly of an isolate of the soil-born phytopathogenic fungus Rosellinia necatrix. Based on effector prediction, the authors identified a set of candidates with potential antimicrobial activity. They demonstrate experimentally antifungal and antibacterial action for two of these candidates. The authors speculate that these effectors contribute to fungal virulence by interfering with plant-associated beneficial microbiota in the course of infection.

This work nicely combines a high-quality genome assembly of a fungal pathogen with functional implications derived from its genome sequence. Overall, it is an interesting case study, further promoting the idea of molecular warfare between fungal pathogens and plant microbiota. However, I feel that the manuscript would benefit from some improvements, especially additional controls to strengthen the main claims of this report.

Reviewer #2: Chavarro-Carrero et al. present their work on the sequencing and assembly of the genome of Rosellinia necatrix, the causal agent of white root rot disease in several plant species. The team sequenced nine strains of R. necatrix using long-read sequencing (ONT) and also used chromosome conformational capture (Hi-C) on the strain R18 to make a high-quality reference for the pathogen. They use standard pipeline for gene annotation. Identification of effector content found no evidence of a two-speed genome, as previously observed for other plant pathogens. In parallel, the genome exhibited low occurrence of repetitive sequence. Next, they clustered effectors based on their predicted structure using AlphaFold2. They found that effectors formed small clusters that lacked evidence of expansion (as observed in other plant pathogens). Based on previous work from the group on the interaction of root fungal plant pathogens and soil microbiome, the team investigated whether culture filtrate of R. necatrix harbored microbial activity. A total of 1 of 37 plant-associated bacteria were inhibited from multiple time points of filtrate and another three species were inhibited at a single time point. To identify candidate effectors that may contribute to this and other antimicrobial activity, the team identified candidate secreted proteins, attempted heterologous expression in E. coli (they were successful with one protein, FUN_004580), and tested direct activity on different fungal species. A second protein, FUN_011519, was generated using protein synthesis. Based on structural similarity to other proteins with antifungal activity, they tested these proteins against a collection of diverse fungi (filamentous and yeasts). FUN_004580 exhibited broad spectrum antifungal activity, whereas FUN_011519 showed selective antifungal activity. They also observed selective antibacterial activity of both proteins. Using antagonistic assays between R. necatrix, they found overlap between the protein-bacterial and R. necatrix-bacterial assays, with some additional antagonistic activity. Causality of these antimicrobial proteins was not demonstrated using mutants in R. necatrix. Lastly, the team investigated the impact of inoculating different bacterial species as a protective assay to R. necatrix infection of cotton. They found that the four tested plant-associated bacteria all contributed a protective ability to cotton.

Strengths

State-of-the-art methods for genome assembly and annotation

Extensive testing of antagonistic activity of diverse bacterial and fungal species to R. necatrix

Correlation of antimicrobial activity of specific peptides with R. necatrix antimicrobial activity

Weakness

Loss-of-function mutants in R. necatrix not produced for FUN_004580 and FUN_011519

Reviewer #3: The authors have sequenced and annotated the genome of the soil-borne white root rot pathogen Rosellinia necatrix. They have obtained sequences of different host-adapted strains of different countries and were able to completely assemble the nuclear and mitochondrial genomes of strain R18. While analyzing the effector content, they did not find signs of genome compartmentalization and only weak structural clustering of effector candidates. Culture filtrates of R. necatrix had antimicrobial activity, and several peptides with putative antimicrobial activity could be predicted. Of these, two were further studied and found to have structural similarity to known antifungal proteins. They assessed their antimicrobial activity and found activity against distinct fungal and bacterial species, that were found to reduce fungal virulence in plant co-infection experiments.

The manuscript is of very high quality, excellently prepared and well-written. The story is highly interesting and complete, and all conclusions are backed up by experimental data. It fits perfectly with the scope of the journal.

**Part II – Major Issues: Key Experiments Required for Acceptance**

Reviewer #1: 1.) Figure 5: In addition to the heat inactivation, it would be desirable to see whether the antimicrobial activity of the culture filtrate is also protease-sensitive. This could be tested with a selected number of bacteria that are inhibited in growth by the culture filtrate (e.g. B. drentensis, A. denitrificans and S. mellinum).

2.) Figure 9: As for Figure 5, I would request to test for the heat- and protease-sensitivity of the antimicrobial effect of the two selected R. necatrix effectors using one or two representative fungal/bacterial cases as additional controls.

3.) Figure 11: Similar to my comments to Figures 5 and 9, I wonder what the effect of pre-treatment of the cotton seeds with heat-killed bacteria would be. Also, as an additional control, preincubation with bacteria that likely would not inhibit R. necatrix would be desired. In other words: Is the bacterial inhibition shown in this Figure an effect specific for the bacteria chosen or would any bacterium inhibit growth of the fungal phytopathogen?

Reviewer #2: 1. The authors make the claim that the genome does not appear to experience a two-speed genome. While this may be true, it appears to be confounded with a very low rate of repetitive sequences in the genome. Perhaps this should be pointed out more strongly, as this result is very different than many other plant pathogen genomes. Has any other group identified a two-speed genome with a plant pathogen with this low of a repeat content?

2. The authors state: “This suggest that geographic origin of the strains correlates with their phylogenetic relationship.” This statement seems to be confounded with host sampling, therefore it seems to have limited scope.

3. “This analysis revealed that almost 40 percent of the effector candidates are structurally unique, while the remaining effectors could be assigned to a total of 31 clusters.” How many of the effector candidates had reasonable folds predicted by AlphaFold2? What was the distribution of proteins in the MSA for these effectors? The unique status may be due to insufficient training data sets for the MSA component of the analysis. Was any clustering performed on the proteins prior to analysis, perhaps with OrthoMCL?

4. Why were candidate antimicrobial peptides not also tested against Rosellinia necatrix as a control?

5. Figure 11. To what extent is this observation simply PTI induction versus direct antagonistic activity? The experiment lacks negative controls.

Reviewer #3: (No Response)

**Part III – Minor Issues: Editorial and Data Presentation Modifications**

Reviewer #1: 1.) Table 1: Could the authors provide BUSCO values for all nine sequenced R. necatrix strains?

2.) Effector prediction: As the authors have available genomic sequence data from nine R. necatrix strains, it would be desirable to see a comparison across these strains regarding conservation of the predicted effectors (both presence/absence and SNPs). Of course there is the caveat of incomplete genome assemblies for eight of the strains, but the quality parameters look rather decent.

3.) Figure 3: I do not know the Realphy tool, but would it be possible to derive bootstrap values or any other measure of reliability of the phylogenetic tree?

4.) Figure 5: It would be good to know on how many replicates/experiments the data shown in Figure 5 are based and what the error bars show. Moreover, I cannot see a blue line for some of the bacteria (e.g. A. denitrificans, but also others). If in these cases the 4-day time point was omitted, this should be indicated in the Figure legend.

5.) Figure 10: I assume the right column of photographs is a negative control of bacterial growth in the absence of R. necatrix? Should be mentioned in the Figure legend. What does n=5 mean here? We only see one set of photographs. If five replicates exist, could the authors provide quantitative data for this experiment by quantifying the area of fungal growth and calculate inhibition?

6.) Figure 11: What does “Mix” mean here? All four bacteria mixed? Should be mentioned in the Figure legend.

7.) Materials and Methods: The origin of the fungal isolates should be more clearly stated. When and where (“Mexico” is a rather broad description) were these collected?

8.) Line 263: “suggest” should read “suggests”

9.) Line 282: Not only fungi, but also oomycetes (e.g. Phytophthora!). Maybe better “filamentous phytopathogens (fungi and oomycetes)

10.) Line 316: “cluster” should read “clusters”

11.) Line 374 ff. - Prediction of antimicrobial effector candidates: What is known about the antimicrobial mechanism of AFP1 and AcAMP2? How do they act? Maybe add to Discussion.

12.) Line 473: “quantify” should read “quantified”

13.) Line 552: Delete “a” in front of “specific”

14.) Line 736: Please provide the R. necatrix gene identifier for the GSAPDH reference gene

15.) Line 997: “Plant Phytologist” should read “New Phytologist”

Reviewer #2: Minor concerns

1. Did you investigate why some BUSCO genes were missing? Were these a subset of proteins that are more variable? Did the annotation pipeline miss genes that are present in the genome?

2. Lines 577-579 “In this study, we show that the soil-borne pathogen Rosellinia necatrix similarly secretes effector proteins during host colonization that possess selective antimicrobial activity.” No experiment was performed to show that these effector proteins were secreted, therefore this claim cannot be made. Throughout, it should be clear that they are predicted secreted proteins unless further evidence is provided.

Minor edits

Author summary: Change “antimicrobials target plant-associated bacterial” to “antimicrobials target plant-associated bacteria”

Figure 2. “sparce” to “sparse”

Table 3. Commas were used in the “Prediction confidence” column.

Lines 381, 383. “asses” to “assess”

Reviewer #3: I have the following minor remarks.

Line 161 explain the abbreviation HMW at first mention

Lines 263-264. I partially disagree with this suggestion since phylogenetic relationship could also correlate with the host plant.

Line 283 Figure 3A. Please consider using better to distinguish colors for Deletion and Translocation

Lines 381 and 383: assess instead of asses

Line 491 and 524 bacterial species

Line 493 Fig. 10 instead of Fig. 9

Line 497 Fig. 10 instead of Fig. 8

Line 509 “the fungus can promote the suppression of disease development” sounds clumsy to me. How about “the fungus can inhibit disease development”?

Lines 513 and 520: Fig. 11 instead of Fig. 10

Line 552 “towards specific”, delete “a”

Lines 561-564 The word complement seems to be used with an unusual meaning for biologists. Normally, if two proteins can complement each other, the one can fulfill the function of the other in the others absence. Here, the activities of the two proteins are clearly different. I suggest rephrasing.

Lines 616-619 In this sentence, I can understand each half sentence, however it is not clear to me, why the second half sentence would result from the first. It would be helpful if the authors could clarify this in the text.

PLOS authors have the option to publish the peer review history of their article (what does this mean?). If published, this will include your full peer review and any attached files.

Reviewer #1: No

Reviewer #2: No

Reviewer #3: **Yes: **Jan Schirawski
---

## [Decision Letter · Decision Letter 1]

20 Nov 2023

Dear Dr. Thomma,

Thank you very much for submitting your manuscript "The soil-borne white root rot pathogen  Rosellinia necatrix  expresses antimicrobial proteins during host colonization" for consideration at PLOS Pathogens. As with all papers reviewed by the journal, your manuscript was reviewed by members of the editorial board and by several independent reviewers. The reviewers appreciated the attention to an important topic. Based on the reviews, we are likely to accept this manuscript for publication, providing that you modify the manuscript according to the review recommendations.

Thank you for addressing all critical comments constructively. A few minor points by reviewer 1 remain. These should be easy to fix in a minor revision.

Sincerely,

Sebastian Schornack, Ph.D.

Academic Editor

PLOS Pathogens

Shou-Wei Ding

Section Editor

PLOS Pathogens

Kasturi Haldar

Editor-in-Chief

PLOS Pathogens

orcid.org/0000-0001-5065-158X

Michael Malim

Editor-in-Chief

PLOS Pathogens

orcid.org/0000-0002-7699-2064

Thank you for addressing all critical comments constructively. A few minor points by reviewer 1 remain. These should be easy to fix in a minor revision.

Reviewer Comments (if any, and for reference):

Reviewer's Responses to Questions

**Part I - Summary**

Reviewer #1: I had a look at the revision and the answers provided by the authors to the various reviewer questions. Although the authors did not address all issues raised by experimental work, I feel that overall they did a good job with their revision and provide plausible arguments why they could not do or refused to do certain experiments. I feel the work improved susbtantially. I just have two minor points (as a consequence of the first revision) that should be addressed.

**Part III – Minor Issues: Editorial and Data Presentation Modifications**

Reviewer #1: 1.) I appreciate that the robustness of the tree shown in Figure 3 was assessed by 1000 bootstrap replicates. Typically, these only in part support the original tree; that is why usually "bootstrap values" (e.g. % support) are given at branches of phylogenetic trees. These should be included in the tree.

2.) The legend of Figure 5B is somewhat unclear. Do the black, light green and pink lines relate to sampling at 4, 7 and 9 days of fungal growth? If so, the legend needs adjustment. A color-coded legend in the Figure would also help readers and is suggested to be included (also true for Figure 9).

PLOS authors have the option to publish the peer review history of their article (what does this mean?). If published, this will include your full peer review and any attached files.

Reviewer #1: No

Figure Files:

Data Requirements:

Reproducibility:

References:

---

## [Editor Report · Decision Letter 2]

27 Nov 2023

Dear Dr. Thomma,

We are pleased to inform you that your manuscript 'The soil-borne white root rot pathogen  Rosellinia necatrix  expresses antimicrobial proteins during host colonization' has been provisionally accepted for publication in PLOS Pathogens.

Best regards,

Sebastian Schornack, Ph.D.

Academic Editor

PLOS Pathogens

Shou-Wei Ding

Section Editor

PLOS Pathogens

Kasturi Haldar

Editor-in-Chief

PLOS Pathogens

orcid.org/0000-0001-5065-158X

Michael Malim

Editor-in-Chief

PLOS Pathogens

orcid.org/0000-0002-7699-2064
---

## [Editor Report · Acceptance letter]

4 Dec 2023

Dear Dr. Thomma,

We are delighted to inform you that your manuscript, "The soil-borne white root rot pathogen *Rosellinia necatrix* expresses antimicrobial proteins during host colonization," has been formally accepted for publication in PLOS Pathogens.

Best regards,

Michael Malim

Editor-in-Chief

PLOS Pathogens

orcid.org/0000-0002-7699-2064